# Swift-FedGNN: Federated Graph Learning with Low Communication and Sample Complexities

## Abstract

Graph neural networks (GNNs) have achieved great success in a wide variety of graph-based learning applications. To expedite training for large-scale graphs, distributed GNN training has been proposed using sampling-based mini-batch training. However, such a traditional distributed GNN training approach is not applicable to emerging GNN learning applications with geo-distributed input graphs, which require the data to be kept within the site where it is generated to protect privacy. On the other hand, federated learning (FL) has been widely used to enable privacy-preserving training under data parallelism. However, because of cross-client links in the aforementioned geo-distributed graph data, applying federated learning directly to GNNs incurs expensive cross-client neighbor sampling and communication costs due to the large graph size and the dependencies between nodes among different clients. To overcome these challenges, we propose a new mini-batch and sampling-based federated GNN algorithmic framework called Swift-FedGNN that primarily performs efficient parallel local training and periodically conducts time-consuming cross-client training. Specifically, in Swift-FedGNN, each client *primarily* trains a local GNN model using only its local graph data, and some randomly sampled clients *periodically* learn the local GNN models based on their local graph data and the dependent nodes across clients. We theoretically establish the convergence performance of Swift-FedGNN and show that it enjoys a convergence rate of $\mathcal{O}\left(T^{-1/2}\right)$, matching the state-of-the-art (SOTA) rate of sampling-based GNN methods, despite operating in the challenging FL setting. Extensive experiments on real-world datasets show that Swift-FedGNN significantly outperforms the SOTA federated GNN approaches with comparable accuracy in terms of efficiency.

## 1 Introduction

**1) Background and Motivation:** Graph neural networks (GNNs) have received increasing attention in recent years and have been widely used across various applications, such as social networks (Deng et al., 2019; Qiu et al., 2018; Wang et al., 2019a), recommendation systems (Ying et al., 2018; Wang et al., 2019b;d), traffic prediction (Cui et al., 2019; Kumar et al., 2019; Li et al., 2019), drug discovery (Wang et al., 2022b; Do et al., 2019; Fout et al., 2017), and disease prediction (Ghorbani et al., 2022; Kazi et al., 2023; Li & Zhang, 2024). GNN learns the high-level graph representations by iteratively aggregating the neighboring features of each node, which is then used for downstream tasks, such as node classification (Kipf & Welling, 2017; Hamilton et al., 2017), link prediction (Yao et al., 2023b; Zhang & Chen, 2018), and graph classification (Zhang et al., 2018; Bacciu et al., 2018).

However, real-world graph datasets can be extensive in scale (*e.g.*, Microsoft Academic Graph (Wang et al., 2020) with over 100 million nodes) and generated in a geo-distributed fashion (Yao et al., 2023a). Similar to traditional datasets (*e.g.*, images), graph datasets may be collected across multiple geo-distributed sites/devices and stored locally. Collecting the entire dataset onto a single site/device not only incurs prohibitively high communication costs, but may also violate data protection regulations. Unlike the traditional datasets where data is mutually independent, the nodes in graph data are usually dependent (shown as the links between nodes in Figure 1). Such large-scale real-world graph datasets often exceed the memory and computational capabilities of a single device (*e.g.*, GPU), which leads to *a compelling need* for developing distributed GNN training with multiple devices or machines (Fey & Lenssen, 2019; Zheng et al., 2020). A common paradigm in distributed GNN training involves subgraph sampling (Zeng et al., 2020) and mini-batch training on each device (Luo et al., 2022). In

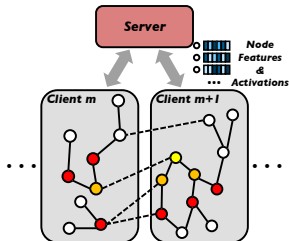

Figure 1: Federated GNN setting. Dashed lines show graph dependency cross-clients.

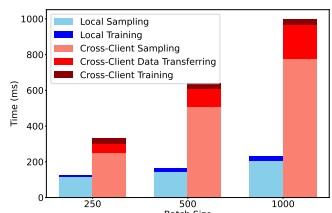

Figure 2: Per-iteration time breakdown: local vs. cross-client training.

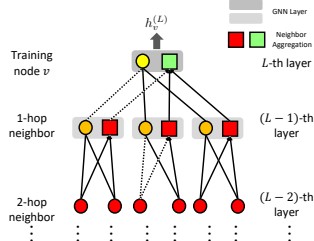

Figure 3: Federated GNN model. Dashed lines show communication between clients.

this approach, workers on each device select training nodes, perform cross-device sampling to gather multi-hop neighbor features as subgraphs, construct subgraphs as mini-batches, and then train on these mini-batches in parallel. However, due to the aforementioned data privacy constraints, this distributed GNN training paradigm is not directly applicable to geo-distributed graphs, as features *cannot* be directly shared among clients.

Meanwhile, federated learning (FL) (McMahan et al., 2017; Yang et al., 2021; Karimireddy et al., 2020), which has emerged as a promising learning paradigm, enables collaborative training of a model using geo-distributed traditional datasets under the coordination of a central server. However, applying FL to geo-distributed graph data is highly non-trivial due to the dependencies between the nodes in a graph and the fact that the neighbors of the node may be located on different clients, which we refer to as *"cross-client neighbors"* (shown as the dashed links between nodes in Figure 1). Ignoring the cross-client neighbors as in (Wang et al., 2022a; He et al., 2021b) would degrade the performance of the models and prevent them from reaching the same accuracy as the models trained on a single device/machine, which is due to the information loss of the cross-client neighbors.

**2) Technical Challenges:** A naive solution to federated GNN is to leverage the server as an intermediary to perform subgraph sampling and part of the training operation (*i.e.*, neighbor aggregation) to protect data privacy. For example, consider a country that has many hospitals and one medical administrative center and needs to investigate a healthcare problem in the whole country (*e.g.*, infection prediction) (Zhang et al., 2021). The residents may go to different hospitals for healthcare because of various reasons, *e.g.*, the locations of the hospitals. Their healthcare data (*e.g.*, personal information, and patient interactions) would be stored locally only at the hospitals they visit, and such data *cannot* be directly shared among different hospitals due to privacy concerns and conflicts of interest. Note that in a graph, the patients here are the nodes and the patient interactions are the edges, and the graphs located at different hospitals may have cross-hospital edges and thus have cross-hospital neighbors. In this situation, the medical administrative center can serve as the server in federated GNN because it is trusted and thus has access to the graph data located at different hospitals.

However, this method introduces significant sampling and communication overhead (as shown in Figure 1), as the server needs to communicate with all clients to perform subgraph sampling and neighbor aggregation for each client sequentially. Figure 2 illustrates the per-iteration time breakdown of local training (*i.e.*, training only on the local graph of the client) versus cross-client training (*i.e.*, training using both the client's local graph and the cross-client neighbors) on the Amazon product co-purchasing dataset (Leskovec et al., 2007) [1]. As observed from the figure, cross-client sampling and data communication time dominate the total time for cross-client training, making it *five times* slower than local training. Therefore, it is critical to mitigate the *communication overhead in* cross-client training to enable efficient federated learning on GNNs.

While some prior works ignore the information of the cross-client neighbors (He et al., 2021a) or assume overlapping nodes between different clients (Wu et al., 2021) (may not hold for geo-distributed graphs), other prior works (Zhang et al., 2021; Du & Wu, 2022; Yao et al., 2023a) address the information loss of the cross-client neighbors by facilitating the exchange of such information between clients. However, these approaches may lead to significant sampling and communication

---

[1]Figure 2 uses a two-layer GNN, with sampling fanout values being 15 and 10 for the two layers, and the network bandwidth being 1 Gbps. We ignore model synchronization time, as the model (of size 0.3 MB) takes less than 10 ms to synchronize.

overhead, which is due to cross-client sampling and cross-client neighbor information transferring. In addition, they may impose heavy memory burdens, which is attributed to the information (graph structure and node features) storage of the cross-client neighbors on the clients. (see detailed discussions in Section 2).

**3) Our Contributions:** The key contribution of this paper is that, by addressing the above challenges, we develop a mini-batch-based and sampling-based federated GNN framework called Swift-FedGNN. The main results and technical contributions of this paper are as follows:

- We develop a new communication- and sample-efficient mini-batch sampling-based federated GNN algorithm called Swift-FedGNN to train GNNs on geo-distributed graphs in a federated fashion. To reduce the sampling and communication overhead, the clients in Swift-FedGNN *primarily* conduct the efficient local training in parallel and some sampled clients *only occasionally and periodically* perform the time-consuming cross-client training. The information loss of the cross-client neighbors in the federated setting is alleviated via cross-client training. Thanks to our use of different mini-batch training nodes at each iteration, the clients do *not* need to store the cross-client neighbor information, which significantly reduces the memory overhead. To further reduce the communication cost, the cross-client neighbor information is aggregated at remote clients before communicating to the server and accumulated one more time before transferring to the training client. This special design further offers the benefit of helping preserve data privacy since the information of each node is not leaked.

- We conduct rigorous theoretical convergence performance analysis for Swift-FedGNN. It is worth noting that the convergence analysis of our Swift-FedGNN is highly non-trivial. Unlike deep neural networks (DNNs), the stochastic gradients in GNNs are *biased*, which poses significant challenges on the theoretical analysis of Swift-FedGNN's convergence guarantees. Moreover, the structural entanglement in GNNs (*i.e.*, the interleaving of neighbor aggregations and non-linear transformations across multiple layers) further complicates the performance analysis. In stark contrast to existing works in the literature that made strong assumptions on the biases of stochastic gradients (e.g., the unbiased stochastic gradient assumption in (Chen et al., 2018) and the consistent stochastic gradient assumption in (Chen & Luss, 2018), etc.), for the *first time* in the literature, we are able to *bound* the stochastic gradient approximation errors rather than resorting to these unrealistic assumptions in practice. Such results could also be of independent theoretical interests.

- Given the *biased* stochastic gradients in GNNs that arise from the missing cross-client neighbors and the neighbor sampling process, we reveal an interesting theoretical insight that the stochastic gradient approximation errors are correlated with the structure of GNNs. More specifically, our theoretical analysis quantifies and characterizes a *positive correlation* with the number of layers in the networks. We note that this is a new finding that is unique to federated GNN training. Lastly, by putting the above insights together, we show that Swift-FedGNN achieves a convergence rate of $\mathcal{O}\left(T^{-1/2}\right)$, which *matches* the state-of-the-art (SOTA) convergence rate of sampling-based GNN methods (hence low communication and sample complexities), *despite* operating in the far more challenging FL setting with much less frequent information exchanges among the clients.

## 2 RELATED WORK

In this section, we provide an overview on distributed GNNs and offer a comprehensive comparison with the most relevant work on federated GNNs.

**1) Distributed Graph Neural Networks:** Distributed GNN training framework (*e.g.*, DGL's Dist-DGL (Wang et al., 2019c; Zheng et al., 2020), Pytorch Geometric (Fey & Lenssen, 2019), Ali-Graph (Zhao et al., 2019) and Dorylus (Thorpe et al., 2021)) have been developed to train large-scale graph datasets that exceed the storage capacities of a single device. Each worker on the device constructs mini-batches via cross-device sampling and communication, trains in parallel, and synchronizes the model. However, in distributed GNN training, extensive graph sampling and data communication can account for up to 80% of the total training time, substantially slowing the training process (Gandhi & Iyer, 2021). Considerable efforts have been made to optimize distributed GNN training, including employing strategic graph partitioning to minimize edge cuts between graph partitions (Zheng et al., 2020), implementing static or dynamic node feature caching (Liu et al., 2023; Zhang et al., 2023), enhancing communication strategies (Cai et al., 2021; Luo et al., 2022), and utilizing various parallel training schemes (Gandhi & Iyer, 2021; Wan et al., 2022; Du et al., 2024). However, the majority of these techniques are not directly applicable to geo-distributed graphs due

to privacy concerns, as they typically involve operations that require access to or the transfer of graph data across different training devices. To our knowledge, LLCG (Ramezani et al., 2022) is the only distributed GNN training framework that avoids transferring node features between workers, making it potentially applicable to geo-distributed graphs. Every worker in LLCG trains only on its local graph partitions. To address the missing information from cross-device neighbors, LLCG employs central server to periodically perform full-neighbor training with neighbor aggregation over all workers. However, this method suffers significant communication overhead on the server end, as the server needs to communicate with all workers to perform the full-neighbor training.

**2) Federated Graph Neural Networks:** To date, the research on federated GNNs remains in its infancy and results in this area are quite limited. In (He et al., 2021a), it is assumed that graphs are dispersed across multiple clients and the information of the cross-client neighbors is ignored, which does not align with the real-world scenarios and would degrade the performance of the trained model. In (Wu et al., 2021), it is assumed that the clients' local graphs have overlapped nodes and the edges are distributed, which may not be true in real-world situations. The authors in (Zhang et al., 2021) mitigate the information loss of the cross-client neighbors by exchanging such information in each training round. However, this approach incurs considerable communication overhead and exposes private node information to other clients. Although the algorithm in (Yao et al., 2023a) exchanges the information of the full cross-client neighbors only once before the training to supplement the missing information from the cross-client neighbors, it is not applicable to large-scale graphs because it uses the full graph for training and incurs significant memory overhead on each client. Note that this one-time communication only works for full graph training and is not suitable for the situations in which clients sample different mini-batches of training nodes in each iteration since the cross-client neighbors of these training batches are different. Furthermore, the homomorphic encryption used in (Yao et al., 2023a) would significantly increase the communication cost, which is at least several times higher than communication without homomorphic encryption.

The most related work to ours can be found in (Du & Wu, 2022), where the authors used sparse cross-client neighbor sampling to supplement the lost information of the cross-client neighbors and reduce the communication overhead. Each client periodically samples the cross-client neighbors and exchanges the information of the sampled neighbors with other clients. In the remaining iterations, the clients reuse the most recent sampled cross-client neighbors, which requires additional cache for saving transferred graph data, thereby increasing memory overhead. However, as training progresses, the frequency of information exchange increases, leading to higher communication costs. Additionally, they relaxed the privacy constraint to allow the transfer of the graph data between clients directly. Reusing the same sampled data across multiple iterations would cause additional bias, and thus degrading the performance of the model. In contrast, our proposed Swift-FedGNN method limits cross-client training to a *subset* of sampled clients and avoids direct graph data exchange between clients by offloading certain operations to the central server. Before communication with the training clients, cross-client neighbor information is aggregated twice: first at the remote clients and then on the server—helping to preserve data privacy and significantly reduce communication costs. Since clients perform local training in the remaining iterations, cross-client neighbor information does *not* need to be stored, thereby reducing memory overhead.

## 3 FEDERATED GRAPH LEARNING: PRELIMINARIES

In this section, we provide the background of the mathematical formulation for training GNNs in a federated setting. For convenience, we provide a list of key notations used in this paper in Appendix A. In order for this paper to be self-contained and to facilitate easy comparisons, we provide the background for training GNNs on a single machine in Appendix B.

Consider a graph $\mathcal{G}\left(\mathcal{V}, \mathcal{E}\right)$, where $\mathcal{V}$ is a set of nodes with $N = |\mathcal{V}|$ and $\mathcal{E}$ is a set of edges. We consider a standard federated setting that has a central server and a set of $\mathcal{M}$ clients with $M = |\mathcal{M}|$. The graph $\mathcal{G}$ is geographically distributed over these clients, and each client $m$ contains a subgraph represented by $\mathcal{G}^m\left(\mathcal{V}^m, \mathcal{E}^m\right)$. Note that $\bigcup_{m=1}^{M} \mathcal{G}^m \neq \mathcal{G}$ due to the missing cross-client edges between clients ($\bigcup_{m=1}^{M} \mathcal{E}^m \neq \mathcal{E}$). In addition, we assume that the nodes are *disjointly* partitioned across clients, *i.e.*, $\bigcup_{m=1}^{M} \mathcal{V}^m = \mathcal{V}$ and $\bigcap_{m=1}^{M} \mathcal{V}^m = \emptyset$. Each node $v \in \mathcal{V}^m$ has a feature vector $\boldsymbol{x}_v^m \in \mathbb{R}^d$, and each node $v \in \mathcal{V}_{train}^m$ corresponds to a label $y_v^m$, where $\mathcal{V}_{train}^m \subseteq \mathcal{V}^m$.

In federated GNN training, the clients collaboratively learn a model with distributed graph data and under the coordination of the central server. Typically, clients in FL receive the model from the server, compute local model updates iteratively, and then send the updated model to the server. The server periodically aggregates the models and then sends the aggregated model back to the clients. The goal in federated GNN training is to solve the following optimization problem:

$$\min \mathcal{L}(\boldsymbol{\theta}) := \frac{1}{|\mathcal{M}|} \sum_{m \in \mathcal{M}} F^m(\boldsymbol{\theta}) = \frac{1}{|\mathcal{M}|} \sum_{m \in \mathcal{M}} \frac{1}{|\mathcal{V}_B^m|} \sum_{v \in \mathcal{V}_B^m} \ell^m \left( \boldsymbol{h}_v^{(L),m}, y_v^m \right), \qquad (1)$$

where $\ell^m$ is a loss function (*e.g.*, cross-entropy loss) at client $m$, $\mathcal{V}_B^m$ denotes a mini-batch of training nodes uniformly sampled from $\mathcal{V}^m$, and $\boldsymbol{\theta} := \left\{ \boldsymbol{W}^{(l)} \right\}_{l=1}^L$ corresponds to all model parameters.

GNNs aim to generate representations (embeddings) for each node in the graph by combining information from its neighboring nodes. Recall that in federated GNNs, the neighbors of node $v$ may be located on its local client $m(v)$ or on remote clients $\bar{m}(v) \in \bar{\mathcal{M}}(v)$, where $\bar{\mathcal{M}}(v)$ represents a set of the remote clients that host the neighbors of node $v$, and $\bar{\mathcal{M}}(v) \subseteq \mathcal{M} \setminus \{m(v)\}$. As shown in Figure 3, to compute the embedding of node $v$ at the $l$-th layer in a GNN with $L$ layers, the client $m(v)$ first aggregates the neighbor information from both itself and the remote clients $\bar{m}(v) \in \bar{\mathcal{M}}(v)$, and then updates the embedding of node $v$, as follows:

$$\boldsymbol{h}_{\mathcal{N}(v)}^{(l)} = \text{AGG}\left( \left\{ \boldsymbol{h}_u^{(l-1),m(v)} \mid u \in \mathcal{N}^{m(v)}(v) \right\} \cup \left\{ \bigcup_{\bar{m}(v) \in \bar{\mathcal{M}}(v)} \left\{ \boldsymbol{h}_u^{(l-1),\bar{m}(v)} \mid u \in \mathcal{N}^{\bar{m}(v)}(v) \right\} \right\} \right),$$

$$\boldsymbol{h}_v^{(l),m(v)} = \sigma\left( \boldsymbol{W}^{(l)} \cdot \text{COMBINE}\left( \boldsymbol{h}_v^{(l-1),m(v)}, \boldsymbol{h}_{\mathcal{N}(v)}^{(l)} \right) \right), \qquad (2)$$

where $\mathcal{N}^{m(v)}(v)$ is a set of the neighbors of node $v$ located on its local client $m(v)$, $\mathcal{N}^{\bar{m}(v)}(v)$ is a set of the neighbors of node $v$ located on remote client $\bar{m}(v)$, $\boldsymbol{h}_{\mathcal{N}(v)}^{(l)}$ is the aggregated embedding from node $v$'s neighbors, $\boldsymbol{h}_v^{(l),m(v)}$ is the embedding of node $v$ located on client $m(v)$ and is initialized as $\boldsymbol{h}_v^{(0),m(v)} = \boldsymbol{x}_v^{m(v)}$, $\boldsymbol{W}^{(l)}$ represents the weight matrix at $l$-th layer, $\sigma(\cdot)$ corresponds to an activation function (*e.g.*, ReLU), $\text{AGG}(\cdot)$ is an aggregation function (*e.g.*, mean), and $\text{COMBINE}(\cdot)$ is a combination function (*e.g.*, concatenation). Compared to distributed GNNs where clients can directly transfer node features, the *key difference* in federated GNNs is that clients *cannot* do so due to privacy concerns, requiring additional modifications.

## 4 THE Swift-FedGNN ALGORITHM

In this section, we propose a new algorithmic framework called Swift-FedGNN , designed to efficiently solve Problem (1) by reducing both sampling and communication costs in federated GNN training. The overall algorithmic framework of Swift-FedGNN is illustrated in Algorithms 1-3. Rather than each client performing cross-client training in every round, the clients in Swift-FedGNN primarily conduct the *efficient* local training in parallel, and a set of randomly selected clients periodically carry out the time-consuming cross-client training. By offloading part of the graph operation to the server and remote clients, Swift-FedGNN eliminates the need for sharing graph features among clients.

Algorithm 1 outlines the main framework of Swift-FedGNN. Specifically, it performs parallel local training across clients for every $I - 1$ iterations, followed by one iteration of cross-client training involving randomly selected clients. In the local training iterations ($t$), every client $m$ updates the local GNN model only using its local graph, as presented in Algorithm 3. Client $m$ samples a mini-batch of training nodes $\mathcal{B}_v^m$ and a subset of $L$-hop neighbors for the training nodes in $\mathcal{B}_v^m$, denote as $\widetilde{\mathcal{S}} = \left\{ \widetilde{\mathcal{S}}^{(l)} \right\}_{l=0}^{L-1}$, all from the local graph data. To compute the embedding of node $v$ in the $l$-th GNN layer ($v \in \mathcal{B}_v^m$ if $l = L$, otherwise $v \in \widetilde{\mathcal{S}}^{(l)}$), client $m$ first conducts the neighbor aggregation for node $v$ based on the sampled neighbors using:

$$\widetilde{\boldsymbol{h}}_{\mathcal{N}(v)}^{(l)} = \text{AGG}\left( \left\{ \widetilde{\boldsymbol{h}}_u^{(l-1),m} \mid u \in \widetilde{\mathcal{N}}^m(v) \right\} \right), \qquad (3)$$

where $\widetilde{\mathcal{N}}^m(v)$ denotes a set of the sampled neighbors located on client $m$ for node $v$, $\widetilde{\mathcal{N}}^m(v) \subseteq \widetilde{\mathcal{S}}^{(l-1)}$, and $\widetilde{\mathcal{N}}^m(v) \subseteq \mathcal{N}^m(v)$. Then, client $m$ updates the embedding of node $v$ in the $l$-th GNN layer based on the aggregated neighbor information and the embedding of node $v$ from the $(l-1)$-th layer, as follows:

**Algorithm 1: Swift-FedGNN Algorithm.**

**Input:** Initial parameters $\boldsymbol{\theta}_0$, learning rate $\alpha$, and correction frequency $I$
**for** $t = 0$ **to** $T - 1$ **do**
    **if** $t \bmod I = 0$ **then**
        Randomly sample $|\mathcal{K}|$ clients
        **for** $m \in \mathcal{M}$ **in parallel do**
            **if** $m \in \mathcal{K}$ **then**
                Client update with local graph data and cross-client neighbors using Algorithm 2
            **else**
                Client update with local graph data according to Algorithm 3
    **else**
        **for** $m \in \mathcal{M}$ **in parallel do**
            Client update with local graph data based on Algorithm 3
    **Server:**
    Aggregate and update global model parameter as:
$$\boldsymbol{\theta}_{t+1} = \boldsymbol{\theta}_t - \alpha \frac{1}{|\mathcal{M}|} \sum_{m \in \mathcal{M}} \nabla \widetilde{F}^m \left( \boldsymbol{\theta}_t^m \right)$$

**Algorithm 2:** Client $m$ in the $t$-th iteration: update with local graph data and cross-client neighbors.

Receive global parameter $\boldsymbol{\theta}_t^m = \boldsymbol{\theta}_t$
Construct a mini-batch $\mathcal{B}_v^m$ of nodes
Server samples a subset of $L$-hop neighbors $\widetilde{\mathcal{S}} = \left\{ \widetilde{\mathcal{S}}^{(l)} \right\}_{l=0}^{L-1}$ for the training nodes in $\mathcal{B}_v^m$
**for** $l = 1$ **to** $L$ **do**
    /* Derive $l$-th layer embedding of node $v \in \mathcal{B}_v^m$ if $l = L$, otherwise $v \in \widetilde{\mathcal{S}}^{(l)}$ */
    **for** *Remote client* $\widetilde{m}(v) \in \widetilde{\mathcal{M}}(v)$ **in parallel do**
        Aggregate the neighbor embeddings using Eq. (5)
        Send the aggregated embedding $\widetilde{\boldsymbol{h}}_{\mathcal{N}(v)}^{(l),\widetilde{m}(v)}$ to the server
    **Server:**
    Aggregate the neighbor embeddings from the remote clients using Eq. (6)
    Send the aggregated cross-client neighbor embedding $\widetilde{\boldsymbol{r}}_{\mathcal{N}(v)}^{(l)}$ to Client $m(v)$
    **Client** $m(v)$**:** Compute node embeddings using Eq. (7) and (8)

Compute the stochastic gradient as $\nabla \widetilde{F}^m \left( \boldsymbol{\theta}_t^m \right)$ and send to the server

**Algorithm 3:** Client $m$ in the $t$-th iteration: update with local graph data.

Receive global parameter $\boldsymbol{\theta}_t^m = \boldsymbol{\theta}_t$
Construct a mini-batch $\mathcal{B}_v^m$ of nodes
Sample a subset of $L$-hop neighbors $\widetilde{\mathcal{S}} = \left\{ \widetilde{\mathcal{S}}^{(l)} \right\}_{l=0}^{L-1}$ for the training nodes in $\mathcal{B}_v^m$
**for** $l = 1$ **to** $L$ **do**
    /* Derive $l$-th layer embedding of node $v \in \mathcal{B}_v^m$ if $l = L$, otherwise $v \in \widetilde{\mathcal{S}}^{(l)}$ */
    Compute node embeddings using Eq. (3) and (4)
Compute the stochastic gradient $\nabla \widetilde{F}^m \left( \boldsymbol{\theta}_t^m \right)$ and send to the server

$$\widetilde{\boldsymbol{h}}_v^{(l),m} = \sigma \left( \boldsymbol{W}_t^{(l),m} \cdot \text{COMBINE} \left( \widetilde{\boldsymbol{h}}_v^{(l-1),m}, \widetilde{\boldsymbol{h}}_{\mathcal{N}(v)}^{(l)} \right) \right). \tag{4}$$

At every $I$-th iteration, Swift-FedGNN allows a set of $K$ clients, uniformly sampled from $\mathcal{M}$, to conduct cross-client training that trains the local GNN models using both their local graph data and the cross-client neighbors. We use $\mathcal{K}$ to denote the set of $K$ clients, where $\mathcal{K} \subset \mathcal{M}$. The remaining clients perform local training as shown in Algorithm 3. Algorithm 2 details the cross-client training process for client $m \in \mathcal{K}$. Rather than directly exchanging node features between clients, Swift-FedGNN partitions GNN training between the clients and the server. We offload[2] the aggregation of node features and intermediate activations at each GNN layer to the server and remote clients corresponding to node $v$, thus reducing the communication overhead and eliminating the need for graph data sharing. This procedure helps preserve data privacy because the clients are unaware of the locations of neighbor nodes, and the embeddings of these neighbor nodes are aggregated before being transmitted to the clients. Operations performed on the server and the remote clients are colored using `server` and `remote client` respectively.

Specifically, client $m \in \mathcal{K}$ samples a mini-batch of training nodes $\mathcal{B}_v^m$. Then, with the cooperation of the server, a subset of $L$-hop neighbors for the training nodes in $\mathcal{B}_v^m$ is sampled and represented as

---

[2]Note that the operation offloading in Swift-FedGNN only supports element-wise (*e.g.*, mean, sum, max) operations, *e.g.*, GCN Kipf & Welling (2017) and SGCN Wu et al. (2019). To support non-element-wise operation, *e.g.*, GAT Veličković et al. (2017), each remote client can transfer the raw graph features or activations to the server for aggregation, instead of performing the locally partial aggregation first with Eq. (5)

$\widetilde{\mathcal{S}} = \left\{ \widetilde{\mathcal{S}}^{(l)} \right\}_{l=0}^{L-1}$. The nodes $v \in \mathcal{B}_v^m$ are on client $m$, while for $v \in \widetilde{\mathcal{S}}^{(l)}$ with $l < L$, the nodes may be on clients other than $m$, denoting the client storing $v$ as $m(v)$. The set $\widetilde{\mathcal{M}}(v)$ represents remote clients with respect to $m(v)$, i.e., $\widetilde{\mathcal{M}}(v) \subseteq \mathcal{M} \setminus \{m(v)\}$, where the sampled cross-client neighbors of the training node $v$ are located. Each remote client $\widetilde{m}(v) \in \widetilde{\mathcal{M}}(v)$ may contain multiple sampled neighbors of the training node $v$, and the numbers of the sampled neighbors can vary across clients.

Computing the $l$-th layer embedding of node $v$ consists of four steps. Steps 1 to 3 below are used to aggregate the neighbor information of node $v$, and Step 4 is used to update the node $v$'s embedding at $l$-th GNN layer.

**Step 1)** Each remote client $\widetilde{m}(v)$ aggregates its sampled neighbors of node $v$ in *parallel,* using

$$\widetilde{\boldsymbol{h}}_{\mathcal{N}(v)}^{(l),\widetilde{m}(v)} = \text{AGG}\left(\left\{\widetilde{\boldsymbol{h}}_u^{(l-1),\widetilde{m}(v)} \mid u \in \widetilde{\mathcal{N}}^{\widetilde{m}(v)}(v)\right\}\right). \tag{5}$$

We send only the aggregated results from each remote client $\widetilde{m}(v)$ to the server, which can help preserve data privacy and reduce communication overhead.

**Step 2)** Upon receiving the aggregated neighbor information from all the remote clients $\widetilde{m}(v) \in \widetilde{\mathcal{M}}(v)$, the server aggregates this information from different remote clients before sending it to client $m(v)$ as follows:

$$\widetilde{\boldsymbol{r}}_{\mathcal{N}(v)}^{(l)} = \text{AGG}\left(\left\{\widetilde{\boldsymbol{h}}_{\mathcal{N}(v)}^{(l),\widetilde{m}(v)} \mid \widetilde{m}(v) \in \widetilde{\mathcal{M}}(v)\right\}\right). \tag{6}$$

This approach not only helps maintain data privacy but also reduces communication costs by minimizing the amount of data transmitted between clients and the server.

**Step 3)** Neighbor information of node $v$ for both the sampled local neighbors and the sampled cross-client neighbors is aggregated as follows:

$$\widetilde{\boldsymbol{h}}_{\mathcal{N}(v)}^{(l)} = \text{AGG}\left(\left\{\widetilde{\boldsymbol{h}}_u^{(l-1),m(v)} \mid u \in \widetilde{\mathcal{N}}^{m(v)}(v)\right\} \cup \left\{\widetilde{\boldsymbol{r}}_{\mathcal{N}(v)}^{(l)}\right\}\right). \tag{7}$$

The cross-client neighbor information used here helps mitigate the information loss and reduce the performance degradation caused by connected nodes being distributed across different clients.

**Step 4)** The embedding of node $v$ in the $l$-th GNN layer is updated using the aggregated neighbor information and the embedding of node $v$ from the $(l-1)$-th layer as:

$$\widetilde{\boldsymbol{h}}_v^{(l),m(v)} = \sigma\left(\boldsymbol{W}_t^{(l),m(v)} \cdot \text{COMBINE}\left(\widetilde{\boldsymbol{h}}_v^{(l-1),m(v)}, \widetilde{\boldsymbol{h}}_{\mathcal{N}(v)}^{(l)}\right)\right). \tag{8}$$

Using the embeddings of the training nodes in the mini-batch and the model parameters, the local stochastic gradients $\nabla \widetilde{F}^m\left(\boldsymbol{\theta}_t^m\right)$ are computed and then used in the update of the global model parameters shown as $\boldsymbol{\theta}_{t+1} = \boldsymbol{\theta}_t - \alpha \frac{1}{|\mathcal{M}|} \sum_{m \in \mathcal{M}} \nabla \widetilde{F}^m\left(\boldsymbol{\theta}_t^m\right)$, where $\alpha$ is the learning rate.

## 5 THEORETICAL PERFORMANCE ANALYSIS

In this section, we establish the theoretical convergence guarantees for Swift-FedGNN using Graph Convolutional Network (GCN) (Kipf & Welling, 2017) as the GNN architecture to solve Problem (1). The analysis of GNN convergence is significantly more challenging compared to the existing literature on deep neural networks (DNNs). The key difficulties stem from the fact that, unlike in DNNs, the stochastic gradients in GNNs are inherently biased. This bias is primarily caused by the presence of cross-client neighbors and the neighbor sampling process. The errors from missing or unsampled neighbors propagate across layers, gradually getting amplified from the input layer to the output layer, complicating the overall convergence behavior.

For a graph $\mathcal{G}$, the structure can be represented by its adjacency matrix $\boldsymbol{A} \in \mathbb{R}^{N \times N}$, where $\boldsymbol{A}_{vu} = 1$ if $(v, u) \in \mathcal{E}$, otherwise $\boldsymbol{A}_{vu} = 0$. The propagation matrix can be computed as $\boldsymbol{P} = \boldsymbol{D}^{-1/2} \hat{\boldsymbol{A}} \boldsymbol{D}^{-1/2}$, where $\hat{\boldsymbol{A}} = \boldsymbol{A} + \boldsymbol{I}$, and $\boldsymbol{D} \in \mathbb{R}^{N \times N}$ corresponds to the degree matrix and $\boldsymbol{D}_{vv} = \sum_u \hat{\boldsymbol{A}}_{vu}$.

For subgraph $\mathcal{G}^m$ located on client $m$, the adjacency matrix $\boldsymbol{A}^m$ can be denoted as $\boldsymbol{A}^m = \boldsymbol{A}_{local}^m + \boldsymbol{A}_{remote}^m$, where $\boldsymbol{A}_{local}^m$ corresponds to the nodes located on client $m$, and $\boldsymbol{A}_{remote}^m$ corresponds to their cross-client neighbors located on the remote clients other than $m$. Then, the propagation matrix

can be calculated as $\boldsymbol{P}^m = \boldsymbol{D}_m^{-1/2} \left( \boldsymbol{A}^m + \boldsymbol{I}^m \right) \boldsymbol{D}_m^{-1/2}$, and can be represented as $\boldsymbol{P}^m = \boldsymbol{P}_{local}^m + \boldsymbol{P}_{remote}^m$, where $\boldsymbol{P}_{local}^m = \boldsymbol{D}_m^{-1/2} \left( \boldsymbol{A}_{local}^m + \boldsymbol{I}^m \right) \boldsymbol{D}_m^{-1/2}$ and $\boldsymbol{P}_{remote}^m = \boldsymbol{D}_m^{-1/2} \left( \boldsymbol{A}_{remote}^m \right) \boldsymbol{D}_m^{-1/2}$.

Given GCN as the GNN architecture, for client $m$ training using only the local graph data, Eq. (3) and (4) are equivalent to $\widetilde{\boldsymbol{H}}_t^{(l),m} = \sigma \left( \widetilde{\boldsymbol{P}}_{local}^{(l),m} \widetilde{\boldsymbol{H}}_{local}^{(l-1),m} \boldsymbol{W}_t^{(l),m} \right)$. For client $m$ training based on both the local graph data and the cross-client neighbors, Eq. (5)–(8) are equivalent to $\widetilde{\boldsymbol{H}}_t^{(l),m} = \sigma \left( \left( \widetilde{\boldsymbol{P}}_{local}^{(l),m} \widetilde{\boldsymbol{H}}_{local}^{(l-1),m} + \widetilde{\boldsymbol{P}}_{remote}^{(l),m} \widetilde{\boldsymbol{H}}_{remote}^{(l-1),m} \right) \boldsymbol{W}_t^{(l),m} \right)$.

Before proceeding with the convergence analysis, we make the following standard assumptions.

**Assumption 5.1.** The loss function $\ell^m \left( \cdot, \cdot \right)$ is $C_l$-Lipschitz continuous and $L_l$-smooth with respect to the node embedding $\boldsymbol{h}^{(L)}$, *i.e.*, $\| \ell^m(\boldsymbol{h}_1^{(L)}, y) - \ell^m(\boldsymbol{h}_2^{(L)}, y) \|_2 \leq C_l \| \boldsymbol{h}_1^{(L)} - \boldsymbol{h}_2^{(L)} \|_2$ and $\| \nabla \ell^m(\boldsymbol{h}_1^{(L)}, y) - \nabla \ell^m(\boldsymbol{h}_2^{(L)}, y) \|_2 \leq L_l \| \boldsymbol{h}_1^{(L)} - \boldsymbol{h}_2^{(L)} \|_2$.

**Assumption 5.2.** The activation function $\sigma \left( \cdot \right)$ is $C_\sigma$-Lipschitz continuous and $L_\sigma$-smooth, *i.e.*, $\| \sigma(\boldsymbol{z}_1^{(l)}) - \sigma(\boldsymbol{z}_2^{(l)}) \|_2 \leq C_\sigma \| \boldsymbol{z}_1^{(l)} - \boldsymbol{z}_2^{(l)} \|_2$ and $\| \nabla \sigma(\boldsymbol{z}_1^{(l)}) - \nabla \sigma(\boldsymbol{z}_2^{(l)}) \|_2 \leq L_\sigma \| \boldsymbol{z}_1^{(l)} - \boldsymbol{z}_2^{(l)} \|_2$.

**Assumption 5.3.** For any $l \in [L]$, the norm of weight matrices, the propagation matrix, and the node feature matrix are bounded by $B_W$, $B_P$ and $B_X$, respectively, *i.e.*, $\| \boldsymbol{W}^{(l)} \|_F \leq B_W$, $\| \boldsymbol{P} \|_F \leq B_P$, and $\| \boldsymbol{X} \|_F \leq B_X$. Note that this assumption is commonly used in the analysis of GNNs, *e.g.*, (Chen et al., 2018; Liao et al., 2020; Garg et al., 2020; Cong et al., 2021; Wan et al., 2022)

Different from DNNs with unbiased stochastic gradients, the stochastic gradients in sampling-based GNNs are *biased* due to neighbor sampling of the training nodes. This is one of the **key challenges** in the convergence performance analysis of Swift-FedGNN. Some existing works used strong assumptions to deal with these biased stochastic gradients in their analysis, *e.g.*, the authors in (Chen et al., 2018) adopted the unbiased stochastic gradient assumption, and the authors in (Chen & Luss, 2018) used the consistent stochastic gradient assumption. However, these assumptions may not hold in reality. In this paper, without using the aforementioned strong assumptions, we are able to bound the errors between the stochastic gradients and the full gradients in the following lemma.

**Lemma 5.4.** *Under Assumptions 5.1–5.3, the errors between the stochastic gradients and the full gradients are bounded as follows:*

$$\left\| \nabla F_{local}^m \left( \boldsymbol{\theta}^m \right) - \nabla \tilde{F}_{local}^m \left( \boldsymbol{\theta}^m \right) \right\|_F \leq L B_{\Delta G}^l, \quad \left\| \nabla F_{full}^m \left( \boldsymbol{\theta}^m \right) - \nabla \tilde{F}_{full}^m \left( \boldsymbol{\theta}^m \right) \right\|_F \leq L B_{\Delta G}^f,$$

*where $\nabla F_{local}^m \left( \boldsymbol{\theta}^m \right)$ and $\nabla \widetilde{F}_{local}^m \left( \boldsymbol{\theta}^m \right)$ correspond to the full and stochastic gradients computed with only the local graph data, respectively. $\nabla F_{full}^m \left( \boldsymbol{\theta}^m \right)$ and $\nabla \widetilde{F}_{full}^m \left( \boldsymbol{\theta}^m \right)$ represent the full and stochastic gradients computed with both the local graph data and the cross-client neighbors of the training nodes, respectively. $B_{\Delta G}^l$ and $B_{\Delta G}^f$ are defined in Eq. (12) and (13) in Appendix D.*

Furthermore, the dependencies of the nodes located on different clients can lead to additional errors in the gradient computations when client $m$ is updated only with its local graph data, since the cross-client neighbors are missed. This becomes another **key challenge** in the analysis of the convergence of Swift-FedGNN. We prove that such an error is upper-bounded as shown in the following lemma.

**Lemma 5.5.** *Under Assumptions 5.1–5.3, the error between the full gradient computed with both the local graph data and the cross-client neighbors of the training nodes (denote as $\nabla F_{full}^m \left( \boldsymbol{\theta}^m \right)$) and the full gradient computed with only the local graph data (denote as $\nabla F_{local}^m \left( \boldsymbol{\theta}^m \right)$) is upper-bounded as follows:*

$$\left\| \nabla F_{full}^m \left( \boldsymbol{\theta}^m \right) - \nabla F_{local}^m \left( \boldsymbol{\theta}^m \right) \right\|_F \leq L B_{\Delta G}^r,$$

*where $B_{\Delta G}^r$ is defined in Eq. (14) in Appendix D.*

We note that all the errors mentioned in Lemmas 5.4 and 5.5 are correlated with the structure of GNNs, specifically showing a positive correlation with the number of layers in the networks. This finding is unique to GNNs, where each layer involves both neighbor aggregation and non-linear transformation. As these two operations are interleaved across multiple layers, they create a structural entanglement that complicates the analysis.

Using Lemmas 5.4 and 5.5, we state the main convergence result of Swift-FedGNN solving an $L$-layer GNN in the following theorem:

**Theorem 5.6.** *Under Assumptions 5.1–5.3, choose step-size* $\alpha = \min\left\{\frac{\sqrt{M}}{\sqrt{T}}, \frac{1}{L_F}\right\}$, *where* $L_F$ *is the smoothness constant in Lemma D.2. The output of* Swift-FedGNN *solving an L-layer GNN satisfies:*

$$\frac{1}{T}\sum_{t=0}^{T-1}\|\nabla\mathcal{L}\left(\boldsymbol{\theta}_t\right)\|^2 \leq \frac{2}{\sqrt{MT}}\left(\mathcal{L}\left(\boldsymbol{\theta}_0\right) - \mathcal{L}\left(\boldsymbol{\theta}^*\right)\right) + L^2\left(B_{\Delta G}^l + B_{\Delta G}^r\right)^2 + \frac{K}{IM}L^2\left(\left(B_{\Delta G}^f\right)^2 - \left(B_{\Delta G}^l + B_{\Delta G}^r\right)^2\right).$$

The detailed proof of Theorem 5.6 can be found in Appendix D. We can see from Theorem 5.6 that the convergence rate of Swift-FedGNN is $\mathcal{O}\left(T^{-1/2}\right)$ to a neighborhood of the exact solution, which *matches* the SOTA convergence rate of sampling-based GNN algorithms, *e.g.*, (Chen et al., 2018; Cong et al., 2021; Ramezani et al., 2022; Du & Wu, 2022), even though Swift-FedGNN operates in the far more challenging federated setting.

Three important remarks on Theorem 5.6 are in order: (1) When choosing $I = 1$ and $K = M$, Swift-FedGNN performs fully cross-client training, ensuring no information loss in the graph data. In this scenario, Swift-FedGNN experiences minimal residual error. Such error is caused by sampling and is inevitable. However, Swift-FedGNN suffers from maximum sampling and communication overhead; (2) When choosing $K = 0$, Swift-FedGNN conducts fully local training, resulting in the information loss of all the cross-client neighbors. Consequently, Swift-FedGNN encounters maximum residual error. Nonetheless, the sampling and communication overhead is minimized; and (3) It can be shown that the last term of the convergence rate bound in Theorem 5.6 is negative. Hence, increasing $I$ or decreasing $K$ would increase the residual error due to more information loss of the cross-client neighbors. However, this would reduce the sampling and communication overhead. Thus, there is a trade-off between the information loss and the sampling and communication overhead.

## 6 NUMERICAL RESULTS

In this section, we conducte experiments to evaluate the performance of Swift-FedGNN.

**1) Experiment Settings:** We train a representative GNN model, GraphSAGE (Hamilton et al., 2017), in the FL settings on two real-world node

Table 1: Benchmark datasets and key parameters.

| DATASET | # OF NODES | # OF EDGES |
|---|---|---|
| OGBN-PRODUCTS | 2.4 M | 61.8 M |
| REDDIT | 0.2 M | 114.6 M |

classification datasets: 1) ogbn-products (Hu et al., 2020), which is an Amazon product co-purchasing graph derived from (Leskovec et al., 2007); and 2) Reddit (Hamilton et al., 2017), which consists of online forum posts within a month, where posts commented on by the same user are connected by an edge. Table 1 summarizes the key statistics of the datasets. Note that Ogbn-products dataset is the *largest* dataset one can find in the federated GNN literature, while the Reddit dataset is known for its *density*. These datasets were chosen for their distinct and representative characteristics, ensuring a thorough evaluation that addresses diverse scenarios in federated GNN training. In our FL simulations, we use 20 clients for the experiments with ogbn-products dataset and 10 clients for the experiments with Reddit dataset. Both graphs are partitioned with METIS partitioning (Karypis & Kumar, 1998). Due to space limitations, additional experimental details and results are provided in Appendix C.

**2) Baselines:** Since the goal of Swift-FedGNN is to reduce the sampling and communication time, we compare Swift-FedGNN with the algorithms most closely related to Swift-FedGNN, which mitigates the information loss of cross-client neighbors through periodical (sampling-based) full-neighbor training: **1)** LLCG (Ramezani et al., 2022): A distributed GNN training framework that performs local training on each client independently, with periodic full-neighbor training conducted on a central server; **2)** FedGNN-PNS (Du & Wu, 2022): A federated GNN training framework where each client periodically samples cross-client neighbors with an increasing sampling frequency. In the remaining iterations, clients reuse the most recently sampled cross-client neighbors; and **3)** FedGNN-G: Naive federated GNN training where cross-client training is performed on each client in every iteration.

**3) Convergence Performance Comparisons:** In Figure 4, we can see that for both the ogbn-products dataset and the Reddit dataset, Swift-FedGNN demonstrates the fastest convergence speed compared to the baseline algorithms, which verifies the effectiveness of Swift-FedGNN. In addition, the validation accuracy of Swift-FedGNN is comparable to that of FedGNN-G, which trains a GNN model on the dataset without any information loss. Specifically, when Swift-FedGNN converges, the validation accuracy is 87.73% on the ogbn-products dataset and 95.60% on the Reddit dataset. When FedGNN-G converges, the validation accuracy is 87.93% on ogbn-products dataset and 96.03% on Reddit dataset. Although LLCG performs periodic cross-client training on the server, it requires

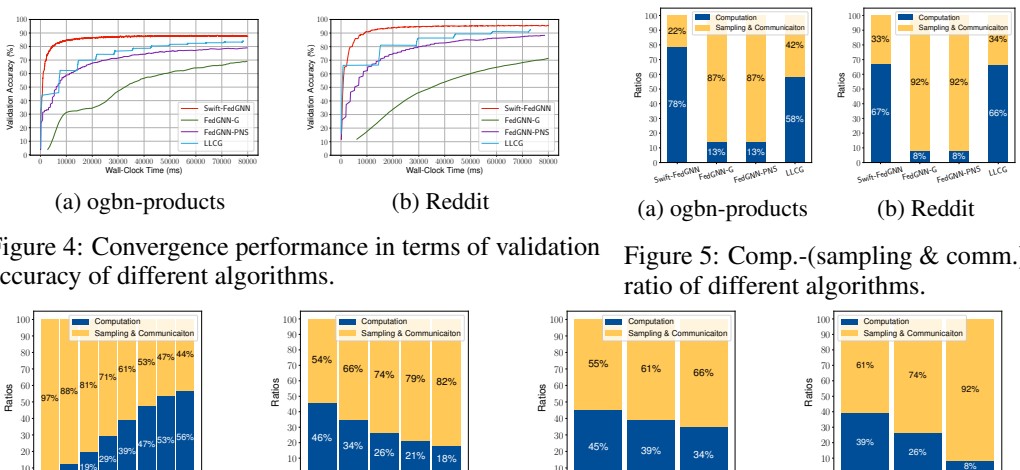

(a) ogbn-products  (b) Reddit

Figure 4: Convergence performance in terms of validation accuracy of different algorithms.

(a) ogbn-products  (b) Reddit

Figure 5: Comp.-(sampling & comm.) ratio of different algorithms.

(a) correction frequencies ($I$) | (b) # of cross-client training clients ($K$) | (c) # of sampled neighbors | (d) # of clients (50% for cross-client training)

Figure 6: Comp.-(sampling & comm.) ratio of Swift-FedGNN on ogbn-products dataset.

training over the full set of neighbors of the training nodes, leading to significant sampling and communication overhead. For instance, when training the ogbn-products dataset, LLCG takes over 5000 ms to perform cross-client training on the server, whereas Swift-FedGNN completes cross-client training within 200 ms due to neighbor sampling. FedGNN-PNS employs a dynamic cross-client sampling interval throughout training, gradually reducing the interval as training progresses. Consequently, FedGNN-PNS incurs extensive sampling and communication overhead during the later stages of training, slowing down the convergence process.

**4) Communication and Sample Costs Analysis:** Figure 5 illustrates the comparison between the ratios of the computation time and the sampling and communication time for Swift-FedGNN and the baseline algorithms. It can be seen that Swift-FedGNN significantly reduces the computation-(sampling & communication) ratio on the ogbn-products dataset. On the Reddit dataset, Swift-FedGNN also significantly reduces this ratio compared to FedGNN-PNS and FedGNN-G. While Swift-FedGNN achieves a comparable ratio to LLCG, it converges much faster and achieves higher validation accuracy than LLCG.

**5) Hyperparameter Sensitivity Analysis:** We explore the impact of the important hyperparameters in Swift-FedGNN. Figure 6a shows that when the correction frequency $I$ increases, the computation-(sampling & communication) ratio increases. Figure 6b and 6c indicate that as the number of cross-client training clients $K$, and the number of sampled neighbors increase, the computation-(sampling & communication) ratio decreases. Figure 6d evaluates Swift-FedGNN with different numbers of clients. In this experiment, 50% of clients periodically conduct cross-client training on both local and cross-client neighbors. We can see that as the number of clients increases, the computation-(sampling & communication) ratio decreases. These findings align with our expectations since sampling and communication overhead is significantly greater than computation overhead in GNN training.

## 7 CONCLUSION

In this paper, we proposed the Swift-FedGNN algorithm, which is a mini-batch-based and sampling-based federated GNN framework, for efficient federated GNN training. Swift-FedGNN reduces the cross-client neighbor sampling and communication overhead by *periodically* sampling a set of clients to conduct the local GNN training on local graph data and cross-client neighbors, which is time-consuming. The rest clients in these periodical iterations and all the clients in the remaining iterations perform efficient parallel local GNN training using only local graph data. We theoretically proved that the convergence rate of Swift-FedGNN is $\mathcal{O}\left(T^{-1/2}\right)$, matching the SOTA rate of sampling-based GNN methods, even in the more challenging federated settings. We conducted extensive numerical experiments on real-world graph datasets and verified the effectiveness of Swift-FedGNN.

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

# A  LIST OF NOTATIONS

| | |
|---|---|
| $\mathcal{G}\left(\mathcal{V},\mathcal{E}\right)$ | Graph |
| $\mathcal{V}$ | Set of nodes |
| $\mathcal{E}$ | Set of edges |
| $N = |\mathcal{V}|$ | Number of nodes |
| $\mathcal{M}$ | Set of clients |
| $M = |\mathcal{M}|$ | Number of clients |
| $\mathcal{G}^m\left(\mathcal{V}^m,\mathcal{E}^m\right)$ | Subgraph at client $m$ |
| $\mathcal{V}^m$ | Set of nodes at client $m$ |
| $\mathcal{E}^m$ | Set of edges at client $m$ |
| $\boldsymbol{x}_v^m \in \mathbb{R}^d$ | Feature vector of node $v$ at client $m$ |
| $y_v^m$ | Label of node $v$ at client $m$ |
| $\ell^m$ | Loss function (*e.g.*, cross-entropy loss) at client $m$ |
| $\mathcal{V}_B^m$ | Mini-batch of training nodes |
| $\boldsymbol{\theta} = \left\{\boldsymbol{W}^{(l)}\right\}_{l=1}^L$ | Set of trainable model parameters |
| $m(v)$ | Local client of node $v$ |
| $\bar{m}(v)$ | Remote client of node $v$ |
| $\bar{\mathcal{M}}(v)$ | Set of the remote clients that host the neighbors of node $v$ |
| $\mathcal{N}^{m(v)}(v)$ | Set of the neighbors of node $v$ located on local client $m(v)$ |
| $\mathcal{N}^{\bar{m}(v)}(v)$ | Set of the neighbors of node $v$ located on remote client $\bar{m}(v)$ |
| $\boldsymbol{h}_v^{(l),m(v)}$ | Embedding of node $v$ located on client $m(v)$ |
| $\boldsymbol{h}_{\mathcal{N}(v)}^{(l)}$ | Aggregated embedding from node $v$'s neighbors |
| $\boldsymbol{W}^{(l)}$ | Weight matrix at $l$-th layer |
| $\sigma\left(\cdot\right)$ | Activation function (*e.g.*, ReLU) |
| $\mathrm{AGG}\left(\cdot\right)$ | Aggregation function (*e.g.*, mean) |
| $\mathrm{COMBINE}\left(\cdot\right)$ | Combination function (*e.g.*, concatenation) |
| $\mathcal{B}_v^m$ | Mini-batch of training nodes at client $m$ |
| $\widetilde{\mathcal{S}} = \left\{\widetilde{\mathcal{S}}^{(l)}\right\}_{l=0}^{L-1}$ | Subset of $L$-hop neighbors for the training nodes in $\mathcal{B}_v^m$ |
| $\widetilde{\mathcal{N}}^m(v)$ | Set of the sampled neighbors located on client $m$ for node $v$ |
| $\mathcal{K}$ | Set of sampled clients for cross-client training |
| $K = |\mathcal{K}|$ | Number of sampled clients for cross-client training |
| $\widetilde{\mathcal{M}}(v)$ | Set of remote clients that host the sampled cross-client neighbors of the training node $v$ |
| $\widetilde{m}(v)$ | Remote client with respect to $m(v)$ |
| $\nabla\widetilde{F}^m\left(\boldsymbol{\theta}_t^m\right)$ | Stochastic gradient |
| $\alpha$ | Learning rate |
| $\boldsymbol{A} \in \mathbb{R}^{N \times N}$ | Adjacency matrix of graph $\mathcal{G}$ |
| $\boldsymbol{P}$ | Propagation matrix |

| | |
|---|---|
| $\boldsymbol{D}$ | Degree matrix |
| $\boldsymbol{A}^m$ | Adjacency matrix of subgraph $\mathcal{G}^m$ |
| $\boldsymbol{A}^m_{local}$ | Adjacency matrix corresponds to the nodes located on client $m$ |
| $\boldsymbol{A}^m_{remote}$ | Adjacency matrix corresponds to the cross-client neighbors located on the remote clients other than $m$ |
| $\boldsymbol{D}^m$ | Degree matrix of client $m$ |
| $\boldsymbol{P}^m$ | Propagation matrix of client $m$ |
| $\boldsymbol{P}^m_{local}$ | Propagation matrix corresponds to the nodes located on client $m$ |
| $\boldsymbol{P}^m_{remote}$ | Propagation matrix corresponds to the cross-client neighbors located on the remote clients other than $m$ |

## B  SINGLE-MACHINE GRAPH NEURAL NETWORKS TRAINING

We consider a graph $\mathcal{G}(\mathcal{V}, \mathcal{E})$, where $\mathcal{V}$ is a set of nodes with $N = |\mathcal{V}|$ and $\mathcal{E}$ is a set of edges. Each node $v \in \mathcal{V}$ is associated with a feature vector $\boldsymbol{x}_v \in \mathbb{R}^d$, where $d$ is the dimension of the feature vector. Each node $v \in \mathcal{V}_{train}$ has a corresponding label $y_v$, where $\mathcal{V}_{train} \subseteq \mathcal{V}$.

GNNs aim to generate representations (embeddings) for each node in the graph by combining information from its neighboring nodes. Consider a GNN that consists of $L$ layers. The embedding of node $v$ at $l$-th layer, which is represented by $\boldsymbol{h}_v^{(l)}$, can be obtained through neighbor aggregation and node update, which are formulated as follows:

$$\boldsymbol{h}_{\mathcal{N}(v)}^{(l)} = \text{AGG}\left(\left\{\boldsymbol{h}_u^{(l-1)} \mid u \in \mathcal{N}(v)\right\}\right), \quad \boldsymbol{h}_v^{(l)} = \sigma\left(\boldsymbol{W}^{(l)} \cdot \text{COMBINE}\big(\boldsymbol{h}_v^{(l-1)}, \boldsymbol{h}_{\mathcal{N}(v)}^{(l)}\big)\right),$$

where $\boldsymbol{h}_v^{(0)}$ is initialized as the feature vector $\boldsymbol{x}_v$, $\mathcal{N}(v)$ denotes the set of neighbors of node $v$, $\boldsymbol{h}_{\mathcal{N}(v)}^{(l)}$ is the aggregated embedding from node $v$'s neighbors aggregated neighbor embedding for node $v$, $\boldsymbol{W}^{(l)}$ represents the weight matrix at $l$-th layer, $\sigma(\cdot)$ corresponds to an activation function (*e.g.*, ReLU), $\text{AGG}(\cdot)$ is an aggregation function (*e.g.*, mean), and $\text{COMBINE}(\cdot)$ is a combination function (*e.g.*, concatenation).

## C  ADDITIONAL EXPERIMENTAL DETAILS AND RESULTS

### C.1  ADDITIONAL EXPERIMENTAL RESULTS

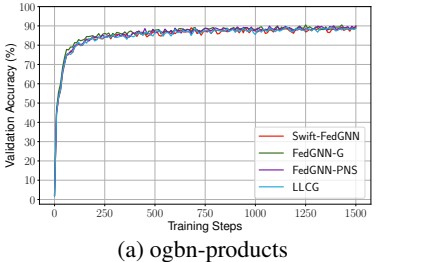
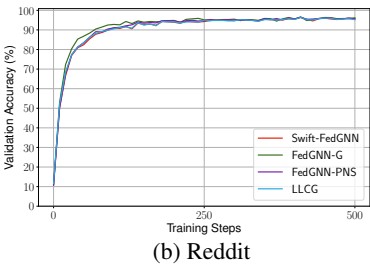

(a) ogbn-products                    (b) Reddit

Figure 7: Convergence performance (validation accuracy versus training steps) of different algorithms.

Table 2: Communication overhead per iteration when communication occurs.

| | Swift-FedGNN | LLCG | FedGNN-PNS | FedGNN-G |
|---|---|---|---|---|
| OGBN-PRODUCTS | 19.5 MB | 378.3 MB | 78.0 MB | 78.0 MB |
| REDDIT | 90.4 MB | 619.6 MB | 180.7 MB | 180.7 MB |

Figure 7 shows that the numbers of iterations required by all algorithms in comparison are similar, which is due to the fact that they share the same convergence rate result. However, since Swift-FedGNN minimizes the sampling and communication overhead, it achieves the lowest wall-clock time for convergence, making it the most efficient in terms of practical implementation.

Table 2 shows the communication overhead per iteration when cross-client sampling and communication occur for different algorithms. We can see that Swift-FedGNN significantly reduces the communication overhead compared to all baselines across both datasets. Specifically, on the ogbn-products dataset, Swift-FedGNN incurs 19.5 MB of overhead per iteration, which is approximately 20 times less than LLCG and 4 times less than both FedGNN-PNS and FedGNN-G. Similarly, for the Reddit dataset, due to its dense inter-node connections and larger feature size, Swift-FedGNN's overhead is 90.4 MB, which is still about 7 times less than LLCG and 2 times less than both FedGNN-PNS and FedGNN-G. This highlights the efficiency of Swift-FedGNN in reducing communication costs during cross-client training.

### C.2 ADDITIONAL EXPERIMENTAL DETAILS

**Implementation and testbed.** We implement Swift-FedGNN using Python on DGL 2.0.0 (Wang et al., 2019c) and PyTorch 2.2.1 (Paszke et al., 2019) with 1241 LoC. Our implementation includes a custom GPU-based sampler built on top of DGL's native sampler, which is designed to sequentially sample local and remote neighbors for each client at every layer. Additionally, we customized the GraphSAGE layer (Hamilton et al., 2017) to facilitate model-parallel training within Swift-FedGNN . In this setup, the server handles the sampling and aggregation of node features and intermediate activations, while the clients are responsible for executing the nonlinear computations associated with the GraphSAGE layer.

We simulate a real-world federated learning scenario using a single machine equipped with NVIDIA Tesla V100 GPUs and 64GB memory. In our setup, both the clients and the server operate on the GPU, and data communication between them is simulated using shared memory. We monitor the data transfer size between the server and clients and set a simulated cross-client network bandwidth at 1Gbps, aligning with real-world measurements reported in (Yuan et al., 2022).

**GNN Model.** We train a two-layer GraphSAGE model with a hidden dimension of 256. Uniform sampling is employed for neighbor sampling, with fan-outs—*i.e.*, the number of sampled neighbors—set according to the official training script provided by the DGL team. The fan-outs for both the ogbn-products dataset and the Reddit dataset are set to be [15, 10]. The training mini-batch size is set at 256. For optimization, we use the Adam optimizer with a learning rate of $0.001$ and a weight decay of $5 \times 10^{-4}$.

## D PROOF OF THEOREM 5.6

### D.1 GRADIENT COMPUTATIONS IN Swift-FedGNN

Recall that Swift-FedGNN uses GCN (Kipf & Welling, 2017) as the architecture of GNN to prove the convergence performance. When client $m$ performs local training that updates the local GNN model using only the local graph data, Each sampling-based GCN layer executes one feature propagation step, defined as:

$$\widetilde{\boldsymbol{H}}_{local}^{(l),m} = \left[ \widetilde{f}^{(l),m} \left( \widetilde{\boldsymbol{H}}_{local}^{(l-1),m}, \boldsymbol{W}^{(l),m} \right) \triangleq \sigma \left( \widetilde{\boldsymbol{P}}_{local}^{(l),m} \widetilde{\boldsymbol{H}}_{local}^{(l-1),m} \boldsymbol{W}^{(l),m} \right) \right].$$

Using the chain rule, the stochastic gradient can be computed as $\nabla \widetilde{F}^m (\boldsymbol{\theta}^m) = \{ \widetilde{\boldsymbol{G}}_{local}^{(l),m} \}_{l=1}^{L}$, where

$$\widetilde{\boldsymbol{G}}_{local}^{(l),m} = \left[ \nabla_W \widetilde{f}^{(l),m} \left( \widetilde{\boldsymbol{D}}_{local}^{(l),m}, \widetilde{\boldsymbol{H}}_{local}^{(l-1),m}, \boldsymbol{W}^{(l),m} \right) \right.$$

$$\triangleq \left[ \widetilde{\boldsymbol{P}}_{local}^{(l),m} \widetilde{\boldsymbol{H}}_{local}^{(l-1),m} \right]^\top \widetilde{\boldsymbol{D}}_{local}^{(l),m} \circ \nabla \sigma \left( \widetilde{\boldsymbol{Z}}_{local}^{(l),m} \right) \right],$$

$$\widetilde{\boldsymbol{D}}_{local}^{(l),m} = \left[ \nabla_H \widetilde{f}^{(l+1),m} \left( \widetilde{\boldsymbol{D}}_{local}^{(l+1),m}, \widetilde{\boldsymbol{H}}_{local}^{(l),m}, \boldsymbol{W}^{(l+1),m} \right) \right.$$

$$\triangleq \left[ \widetilde{\boldsymbol{P}}_{local}^{(l+1),m} \right]^{\top} \widetilde{\boldsymbol{D}}_{local}^{(l+1),m} \circ \nabla \sigma \left( \widetilde{\boldsymbol{Z}}_{local}^{(l+1),m} \right) \left[ \boldsymbol{W}^{(l+1),m} \right]^{\top} \right],$$

in which $\widetilde{\boldsymbol{Z}}_{local}^{(l),m} = \widetilde{\boldsymbol{P}}_{local}^{(l),m} \widetilde{\boldsymbol{H}}_{local}^{(l-1),m} \boldsymbol{W}^{(l),m}$, $\widetilde{\boldsymbol{D}}_{local}^{(L),m} = \partial \ell^m \left( \widetilde{\boldsymbol{H}}_{local}^{(L),m}, \boldsymbol{Y}_{local}^m \right) / \partial \widetilde{\boldsymbol{H}}_{local}^{(L),m}$, and $\circ$ represents Hadamard product.

Similarly, when client $m$ conducts cross-client training that updates the local GNN model based on the local graph data and the cross-client neighbors, each sampling-based GNN layer can be defined as:

$$\widetilde{\boldsymbol{H}}_{full}^{(l),m} = \left[ \widetilde{f}^{(l),m} \left( \widetilde{\boldsymbol{H}}_{full}^{(l-1),m}, \boldsymbol{W}^{(l),m} \right) \triangleq \sigma \left( \left( \widetilde{\boldsymbol{P}}_{local}^{(l),m} \widetilde{\boldsymbol{H}}_{local}^{(l-1),m} + \widetilde{\boldsymbol{P}}_{remote}^{(l),m} \widetilde{\boldsymbol{H}}_{remote}^{(l-1),m} \right) \boldsymbol{W}^{(l),m} \right) \right].$$

Using the chain rule, the stochastic gradient can be calculated as $\nabla \widetilde{F}^m \left( \boldsymbol{\theta}^m \right) = \left\{ \widetilde{\boldsymbol{G}}_{full}^{(l),m} \right\}_{l=1}^{L}$, where

$$\widetilde{\boldsymbol{G}}_{full}^{(l),m} = \left[ \nabla_W \widetilde{f}^{(l),m} \left( \widetilde{\boldsymbol{D}}_{full}^{(l),m}, \widetilde{\boldsymbol{H}}_{full}^{(l-1),m}, \boldsymbol{W}^{(l),m} \right) \right.$$

$$\triangleq \left[ \widetilde{\boldsymbol{P}}_{local}^{(l),m} \widetilde{\boldsymbol{H}}_{local}^{(l-1),m} + \widetilde{\boldsymbol{P}}_{remote}^{(l),m} \widetilde{\boldsymbol{H}}_{remote}^{(l-1),m} \right]^{\top} \widetilde{\boldsymbol{D}}_{full}^{(l),m} \circ \nabla \sigma \left( \widetilde{\boldsymbol{Z}}_{full}^{(l),m} \right) \right],$$

$$\widetilde{\boldsymbol{D}}_{full}^{(l),m} = \left[ \nabla_H \widetilde{f}^{(l+1),m} \left( \widetilde{\boldsymbol{D}}_{full}^{(l+1),m}, \widetilde{\boldsymbol{H}}_{full}^{(l),m}, \boldsymbol{W}^{(l+1),m} \right) \right.$$

$$\triangleq \left[ \widetilde{\boldsymbol{P}}_{local}^{(l+1),m} + \widetilde{\boldsymbol{P}}_{remote}^{(l+1),m} \right]^{\top} \widetilde{\boldsymbol{D}}_{full}^{(l+1),m} \circ \nabla \sigma \left( \widetilde{\boldsymbol{Z}}_{full}^{(l+1),m} \right) \left[ \boldsymbol{W}^{(l+1),m} \right]^{\top} \right],$$

in which $\widetilde{\boldsymbol{Z}}_{full}^{(l),m} = \left( \widetilde{\boldsymbol{P}}_{local}^{(l),m} \widetilde{\boldsymbol{H}}_{local}^{(l-1),m} + \widetilde{\boldsymbol{P}}_{remote}^{(l),m} \widetilde{\boldsymbol{H}}_{remote}^{(l-1),m} \right) \boldsymbol{W}^{(l),m}$, and $\widetilde{\boldsymbol{D}}_{full}^{(L),m} = \partial \ell^m \left( \widetilde{\boldsymbol{H}}_{full}^{(L),m}, \boldsymbol{Y}_{full}^m \right) / \partial \widetilde{\boldsymbol{H}}_{full}^{(L),m}$.

## D.2 USEFUL PROPOSITIONS AND LEMMAS

**Proposition D.1.** *Under Assumption 5.3, the inequalities in Table 3 and Table 4 are hold.*

Table 3: Upper-bound for the norms of the propagation matrix and the node feature matrix.

|  | PROPAGATION MATRIX | NODE FEATURE MATRIX |
|---|---|---|
| FULL GRAPH | $\|\boldsymbol{P}_{full}\|_F \leq B_P$ | $\|\boldsymbol{X}_{full}\|_F \leq B_X$ |
| LOCAL GRAPH | $\|\boldsymbol{P}_{local}\|_F \leq B_P^l \leq B_P$ | $\|\boldsymbol{X}_{local}\|_F \leq B_X^l \leq B_X$ |
| CROSS-CLIENT NEIGHBORS | $\|\boldsymbol{P}_{remote}\|_F \leq B_P^r \leq B_P$ | $\|\boldsymbol{X}_{remote}\|_F \leq B_X^r \leq B_X$ |

Table 4: Relationships for the norms of the propagation matrix and the node feature matrix before and after sampling.

|  | PROPAGATION MATRIX | NODE FEATURE MATRIX |
|---|---|---|
| FULL GRAPH | $\left\|\widetilde{\boldsymbol{P}}_{full} - \boldsymbol{P}_{full}\right\|_F \leq B_{\Delta P}^f$ | $\left\|\widetilde{\boldsymbol{X}}_{full} - \boldsymbol{X}_{full}\right\|_F \leq B_{\Delta X}^f$ |
| LOCAL GRAPH | $\left\|\widetilde{\boldsymbol{P}}_{local} - \boldsymbol{P}_{local}\right\|_F \leq B_{\Delta P}^l$ | $\left\|\widetilde{\boldsymbol{X}}_{local} - \boldsymbol{X}_{local}\right\|_F \leq B_{\Delta X}^l$ |
| CROSS-CLIENT NEIGHBORS | $\left\|\widetilde{\boldsymbol{P}}_{remote} - \boldsymbol{P}_{remote}\right\|_F \leq B_{\Delta P}^r$ | $\left\|\widetilde{\boldsymbol{X}}_{remote} - \boldsymbol{X}_{remote}\right\|_F \leq B_{\Delta X}^r$ |

**Lemma D.2.** *[Lemma 1 in (Cong et al., 2021)] An L-later GCN is $L_F$-Lipschitz smooth, i.e.,* $\|\nabla \mathcal{L}(\boldsymbol{\theta}_1) - \nabla \mathcal{L}(\boldsymbol{\theta}_2)\|_F \leq L_F \|\boldsymbol{\theta}_1 - \boldsymbol{\theta}_2\|_F$.

**Lemma D.3.** *Under Assumptions 5.1–5.3, and for any $l \in [L]$, the Frobenius norm of node embedding matrices, gradient passing from the l-th layer node embeddings to the $(l-1)$-th are bounded, i.e.,*

$$\left\|\boldsymbol{H}_{local}^{(l),m}\right\|_F, \left\|\widetilde{\boldsymbol{H}}_{local}^{(l),m}\right\|_F \leq B_H^l, \qquad \left\|\boldsymbol{H}_{full}^{(l),m}\right\|_F, \left\|\widetilde{\boldsymbol{H}}_{full}^{(l),m}\right\|_F \leq B_H^f,$$

$$\left\|\boldsymbol{D}_{local}^{(l),m}\right\|_F, \left\|\widetilde{\boldsymbol{D}}_{local}^{(l),m}\right\|_F \leq B_D^l, \qquad \left\|\boldsymbol{D}_{full}^{(l),m}\right\|_F, \left\|\widetilde{\boldsymbol{D}}_{full}^{(l),m}\right\|_F \leq B_D^f,$$

*where*

$$B_H^l, B_H^f = \max_{1 \leq l \leq L} \left( C_\sigma B_P B_W \right)^l B_X, \qquad B_D^l, B_D^f = \max_{1 \leq l \leq L} \left( B_P B_W C_\sigma \right)^{L-l} C_l.$$

*Proof.*

$$\left\| \boldsymbol{H}_{local}^{(l),m} \right\|_F = \left\| \sigma \left( \boldsymbol{P}_{local}^{(l),m} \boldsymbol{H}_{local}^{(l-1),m} \boldsymbol{W}^{(l),m} \right) \right\|_F$$

$$\overset{(a)}{\leq} C_\sigma B_W \left\| \boldsymbol{P}_{local}^{(l),m} \boldsymbol{H}_{local}^{(l-1),m} \right\|_F \leq C_\sigma B_W \left\| \boldsymbol{P}_{local}^{(l),m} \right\| \left\| \boldsymbol{H}_{local}^{(l-1),m} \right\|_F$$

$$\overset{(b)}{\leq} C_\sigma B_W B_P \left\| \boldsymbol{H}_{local}^{(l-1),m} \right\|_F \leq (C_\sigma B_W B_P)^l \left\| \boldsymbol{X}^m \right\|_F$$

$$\overset{(c)}{\leq} (C_\sigma B_W B_P)^l B_X \leq \max_{1 \leq l \leq L} (C_\sigma B_W B_P)^l B_X,$$

where (a)–(c) results from Assumptions 5.2 and 5.3.

$$\left\| \widetilde{\boldsymbol{H}}_{local}^{(l),m} \right\|_F = \left\| \sigma \left( \widetilde{\boldsymbol{P}}_{local}^{(l),m} \widetilde{\boldsymbol{H}}_{local}^{(l-1),m} \boldsymbol{W}^{(l),m} \right) \right\|_F$$

$$\overset{(a)}{\leq} C_\sigma B_W \left\| \widetilde{\boldsymbol{P}}_{local}^{(l),m} \widetilde{\boldsymbol{H}}_{local}^{(l-1),m} \right\|_F \leq C_\sigma B_W \left\| \widetilde{\boldsymbol{P}}_{local}^{(l),m} \right\| \left\| \widetilde{\boldsymbol{H}}_{local}^{(l-1),m} \right\|_F$$

$$\overset{(b)}{\leq} C_\sigma B_W B_P \left\| \widetilde{\boldsymbol{H}}_{local}^{(l-1),m} \right\|_F \leq (C_\sigma B_W B_P)^l \left\| \boldsymbol{X}^m \right\|_F$$

$$\overset{(c)}{\leq} (C_\sigma B_W B_P)^l B_X \leq \max_{1 \leq l \leq L} (C_\sigma B_W B_P)^l B_X,$$

where (a)–(c) follow from Assumptions 5.2 and 5.3.

$$\left\| \boldsymbol{H}_{full}^{(l),m} \right\|_F = \left\| \sigma \left( \boldsymbol{P}_{full}^{(l),m} \boldsymbol{H}_{full}^{(l-1),m} \boldsymbol{W}^{(l),m} \right) \right\|_F$$

$$\overset{(a)}{\leq} C_\sigma B_P B_W \left\| \boldsymbol{H}_{full}^{(l-1),m} \right\|_F \leq (C_\sigma B_P B_W)^l \left\| \boldsymbol{X}^m \right\|_F$$

$$\overset{(b)}{\leq} (C_\sigma B_P B_W)^l B_X \leq \max_{1 \leq l \leq L} (C_\sigma B_P B_W)^l B_X,$$

where (a) and (b) are because of Assumptions 5.2 and 5.3.

$$\left\| \widetilde{\boldsymbol{H}}_{full}^{(l),m} \right\|_F = \left\| \sigma \left( \left( \widetilde{\boldsymbol{P}}_{local}^{(l),m} \widetilde{\boldsymbol{H}}_{local}^{(l-1),m} + \widetilde{\boldsymbol{P}}_{remote}^{(l),m} \widetilde{\boldsymbol{H}}_{remote}^{(l-1),m} \right) \boldsymbol{W}^{(l),m} \right) \right\|_F$$

$$\overset{(a)}{\leq} C_\sigma B_W \left\| \widetilde{\boldsymbol{P}}_{local}^{(l),m} \widetilde{\boldsymbol{H}}_{local}^{(l-1),m} + \widetilde{\boldsymbol{P}}_{remote}^{(l),m} \widetilde{\boldsymbol{H}}_{remote}^{(l-1),m} \right\|_F = C_\sigma B_W \left\| \widetilde{\boldsymbol{P}}_{full}^{(l),m} \widetilde{\boldsymbol{H}}_{full}^{(l-1),m} \right\|_F$$

$$\overset{(b)}{\leq} C_\sigma B_W B_P \left\| \widetilde{\boldsymbol{H}}_{full}^{(l-1),m} \right\|_F \leq (C_\sigma B_W B_P)^l \left\| \boldsymbol{X}^m \right\|_F$$

$$\overset{(c)}{\leq} (C_\sigma B_W B_P)^l B_X \leq \max_{1 \leq l \leq L} (C_\sigma B_W B_P)^l B_X,$$

where (a)–(c) follow from Assumptions 5.2 and 5.3.

$$\left\| \boldsymbol{D}_{local}^{(l),m} \right\|_F = \left\| \left[ \boldsymbol{P}_{local}^{(l+1),m} \right]^\top \boldsymbol{D}_{local}^{(l+1),m} \circ \nabla \sigma \left( \boldsymbol{Z}_{local}^{(l+1),m} \right) \left[ \boldsymbol{W}^{(l+1),m} \right]^\top \right\|_F$$

$$\overset{(a)}{\leq} B_W C_\sigma \left\| \boldsymbol{P}_{local}^{(l+1),m} \right\|_F \left\| \boldsymbol{D}_{local}^{(l+1),m} \right\|_F$$

$$\overset{(b)}{\leq} B_P B_W C_\sigma \left\| \boldsymbol{D}_{local}^{(l+1),m} \right\|_F \leq (B_P B_W C_\sigma)^{L-l} \left\| \boldsymbol{D}_{local}^{(L),m} \right\|_F$$

$$\overset{(c)}{\leq} (B_P B_W C_\sigma)^{L-l} C_l \leq \max_{1 \leq l \leq L} (B_P B_W C_\sigma)^{L-l} C_l,$$

where (a)–(c) are because of Assumptions 5.1–5.3.

$$\left\| \widetilde{\boldsymbol{D}}_{local}^{(l),m} \right\|_F = \left\| \left[ \widetilde{\boldsymbol{P}}_{local}^{(l+1),m} \right]^\top \widetilde{\boldsymbol{D}}_{local}^{(l+1),m} \circ \nabla \sigma \left( \widetilde{\boldsymbol{Z}}_{local}^{(l+1),m} \right) \left[ \boldsymbol{W}^{(l+1),m} \right]^\top \right\|_F$$

$$\overset{(a)}{\leq} B_W C_\sigma \left\| \widetilde{\boldsymbol{P}}_{local}^{(l+1),m} \right\|_F \left\| \widetilde{\boldsymbol{D}}_{local}^{(l+1),m} \right\|_F$$

$$\overset{(b)}{\leq} B_P B_W C_\sigma \left\| \widetilde{\boldsymbol{D}}_{local}^{(l+1),m} \right\|_F \leq (B_P B_W C_\sigma)^{L-l} \left\| \widetilde{\boldsymbol{D}}_{local}^{(L),m} \right\|_F$$

$$\overset{(c)}{\leq} (B_P B_W C_\sigma)^{L-l} C_l \leq \max_{1 \leq l \leq L} (B_P B_W C_\sigma)^{L-l} C_l,$$

where (a)–(c) follow from Assumptions 5.1–5.3.

$$\left\| \boldsymbol{D}_{full}^{(l),m} \right\|_F = \left\| \left[ \boldsymbol{P}_{full}^{(l+1),m} \right]^\top \boldsymbol{D}_{full}^{(l+1),m} \circ \nabla \sigma \left( \boldsymbol{Z}_{full}^{(l+1),m} \right) \left[ \boldsymbol{W}^{(l+1),m} \right]^\top \right\|_F$$

$$\overset{(a)}{\leq} B_P B_W C_\sigma \left\| \boldsymbol{D}_{full}^{(l+1),m} \right\|_F \leq (B_P B_W C_\sigma)^{L-l} \left\| \boldsymbol{D}_{full}^{(L),m} \right\|_F$$

$$\overset{(b)}{\leq} (B_P B_W C_\sigma)^{L-l} C_l \leq \max_{1 \leq l \leq L} (B_P B_W C_\sigma)^{L-l} C_l,$$

where (a) and (b) use Assumptions 5.1–5.3.

$$\left\| \widetilde{\boldsymbol{D}}_{full}^{(l),m} \right\|_F = \left\| \left[ \widetilde{\boldsymbol{P}}_{local}^{(l+1),m} + \widetilde{\boldsymbol{P}}_{remote}^{(l+1),m} \right]^\top \widetilde{\boldsymbol{D}}_{full}^{(l+1),m} \circ \nabla \sigma \left( \widetilde{\boldsymbol{Z}}_{full}^{(l+1),m} \right) \left[ \boldsymbol{W}^{(l+1),m} \right]^\top \right\|_F$$

$$= \left\| \left[ \widetilde{\boldsymbol{P}}_{full}^{(l+1),m} \right]^\top \widetilde{\boldsymbol{D}}_{full}^{(l+1),m} \circ \nabla \sigma \left( \widetilde{\boldsymbol{Z}}_{full}^{(l+1),m} \right) \left[ \boldsymbol{W}^{(l+1),m} \right]^\top \right\|_F$$

$$\overset{(a)}{\leq} B_P B_W C_\sigma \left\| \widetilde{\boldsymbol{D}}_{full}^{(l+1),m} \right\|_F \leq (B_P B_W C_\sigma)^{L-l} \left\| \widetilde{\boldsymbol{D}}_{full}^{(L),m} \right\|_F$$

$$\overset{(b)}{\leq} (B_P B_W C_\sigma)^{L-l} C_l \leq \max_{1 \leq l \leq L} (B_P B_W C_\sigma)^{L-l} C_l,$$

where (a) and (b) utilize Assumptions 5.1–5.3.

$\square$

**Lemma D.4.** *Under Assumptions 5.1–5.3, and for any $l \in [L]$, the errors caused by sampling are bounded, i.e.,*

$$\left\| \widetilde{\boldsymbol{H}}_{local}^{(l),m} - \boldsymbol{H}_{local}^{(l),m} \right\|_F \leq B_{\Delta H}^l, \qquad \left\| \widetilde{\boldsymbol{H}}_{full}^{(l),m} - \boldsymbol{H}_{full}^{(l),m} \right\|_F \leq B_{\Delta H}^f,$$

$$\left\| \widetilde{\boldsymbol{D}}_{local}^{(l),m} - \boldsymbol{D}_{local}^{(l),m} \right\|_F \leq B_{\Delta D}^l, \qquad \left\| \widetilde{\boldsymbol{D}}_{full}^{(l),m} - \boldsymbol{D}_{full}^{(l),m} \right\|_F \leq B_{\Delta D}^f,$$

*where*

$$B_{\Delta H}^l = \max_{1 \leq l \leq L} \left( \left( C_\sigma B_W B_H^l B_{\Delta P}^l \right)^l + \left( C_\sigma B_W B_P \right)^l B_{\Delta X}^l \right),$$

$$B_{\Delta H}^f = \max_{1 \leq l \leq L} \left( \left( C_\sigma B_W B_H^f B_{\Delta P}^f \right)^l + \left( C_\sigma B_W B_P \right)^l B_{\Delta X}^f \right),$$

$$B_{\Delta D}^l = \max_{1 \leq l \leq L} \Bigg( \left( B_W B_D^l C_\sigma B_{\Delta P}^l + B_W^2 B_P B_D^l L_\sigma B_H^l B_{\Delta P}^l + B_W^2 B_P^2 B_D^l L_\sigma B_{\Delta H}^l \right)^{L-l}$$

$$+ \left( B_W B_P C_\sigma \right)^{L-l} L_l B_{\Delta H}^l \Bigg),$$

$$B_{\Delta D}^f = \max_{1 \le l \le L} \left( \left( B_W B_D^f C_\sigma B_{\Delta P}^f + B_W^2 B_P B_D^f L_\sigma B_H^f B_{\Delta P}^f + B_W^2 B_P^2 B_D^f L_\sigma B_{\Delta H}^f \right)^{L-l} \right.$$

$$\left. + \left( B_W B_P C_\sigma \right)^{L-l} L_l B_{\Delta H}^f \right).$$

*Proof.*

$$\left\| \widetilde{\boldsymbol{H}}_{local}^{(l),m} - \boldsymbol{H}_{local}^{(l),m} \right\|_F$$

$$= \left\| \sigma \left( \widetilde{\boldsymbol{P}}_{local}^{(l),m} \widetilde{\boldsymbol{H}}_{local}^{(l-1),m} \boldsymbol{W}^{(l),m} \right) - \sigma \left( \boldsymbol{P}_{local}^{(l),m} \boldsymbol{H}_{local}^{(l-1),m} \right) \boldsymbol{W}^{(l),m} \right\|_F$$

$$\overset{(a)}{\le} C_\sigma B_W \left\| \widetilde{\boldsymbol{P}}_{local}^{(l),m} \widetilde{\boldsymbol{H}}_{local}^{(l-1),m} - \boldsymbol{P}_{local}^{(l),m} \boldsymbol{H}_{local}^{(l-1),m} \right\|_F$$

$$\le C_\sigma B_W \left\| \widetilde{\boldsymbol{P}}_{local}^{(l),m} \widetilde{\boldsymbol{H}}_{local}^{(l-1),m} - \boldsymbol{P}_{local}^{(l),m} \widetilde{\boldsymbol{H}}_{local}^{(l-1),m} \right\|_F + C_\sigma B_W \left\| \boldsymbol{P}_{local}^{(l),m} \widetilde{\boldsymbol{H}}_{local}^{(l-1),m} - \boldsymbol{P}_{local}^{(l),m} \boldsymbol{H}_{local}^{(l-1),m} \right\|_F$$

$$\overset{(b)}{\le} C_\sigma B_W B_H^l \left\| \widetilde{\boldsymbol{P}}_{local}^{(l),m} - \boldsymbol{P}_{local}^{(l),m} \right\|_F + C_\sigma B_W B_P \left\| \widetilde{\boldsymbol{H}}_{local}^{(l-1),m} - \boldsymbol{H}_{local}^{(l-1),m} \right\|_F$$

$$\overset{(c)}{\le} C_\sigma B_W B_H^l B_{\Delta P}^l + C_\sigma B_W B_P \left\| \widetilde{\boldsymbol{H}}_{local}^{(l-1),m} - \boldsymbol{H}_{local}^{(l-1),m} \right\|_F$$

$$\le \left( C_\sigma B_W B_H^l B_{\Delta P}^l \right)^l + \left( C_\sigma B_W B_P \right)^l \left\| \widetilde{\boldsymbol{X}}_{local}^m - \boldsymbol{X}_{local}^m \right\|_F$$

$$\overset{(d)}{\le} \left( C_\sigma B_W B_H^l B_{\Delta P}^l \right)^l + \left( C_\sigma B_W B_P \right)^l B_{\Delta X}^l$$

$$\le \max_{1 \le l \le L} \left( \left( C_\sigma B_W B_H^l B_{\Delta P}^l \right)^l + \left( C_\sigma B_W B_P \right)^l B_{\Delta X}^l \right), \tag{9}$$

where (a) uses Assumptions 5.2 and 5.3, (b) is because of Assumption 5.3 and Lemma D.3, and (c) and (d) follow from Proposition D.1.

$$\left\| \widetilde{\boldsymbol{H}}_{full}^{(l),m} - \boldsymbol{H}_{full}^{(l),m} \right\|_F$$

$$= \left\| \sigma \left( \left( \widetilde{\boldsymbol{P}}_{local}^{(l),m} \widetilde{\boldsymbol{H}}_{local}^{(l-1),m} + \widetilde{\boldsymbol{P}}_{remote}^{(l),m} \widetilde{\boldsymbol{H}}_{remote}^{(l-1),m} \right) \boldsymbol{W}^{(l),m} \right) - \sigma \left( \boldsymbol{P}_{full}^{(l),m} \boldsymbol{H}_{full}^{(l-1),m} \right) \boldsymbol{W}^{(l),m} \right\|_F$$

$$\overset{(a)}{\le} C_\sigma B_W \left\| \widetilde{\boldsymbol{P}}_{full}^{(l),m} \widetilde{\boldsymbol{H}}_{full}^{(l-1),m} - \boldsymbol{P}_{full}^{(l),m} \boldsymbol{H}_{full}^{(l-1),m} \right\|_F$$

$$\le C_\sigma B_W \left\| \widetilde{\boldsymbol{P}}_{full}^{(l),m} \widetilde{\boldsymbol{H}}_{full}^{(l-1),m} - \boldsymbol{P}_{full}^{(l),m} \widetilde{\boldsymbol{H}}_{full}^{(l-1),m} \right\|_F + C_\sigma B_W \left\| \boldsymbol{P}_{full}^{(l),m} \widetilde{\boldsymbol{H}}_{full}^{(l-1),m} - \boldsymbol{P}_{full}^{(l),m} \boldsymbol{H}_{full}^{(l-1),m} \right\|_F$$

$$\overset{(b)}{\le} C_\sigma B_W B_H^f \left\| \widetilde{\boldsymbol{P}}_{full}^{(l),m} - \boldsymbol{P}_{full}^{(l),m} \right\|_F + C_\sigma B_W B_P \left\| \widetilde{\boldsymbol{H}}_{full}^{(l-1),m} - \boldsymbol{H}_{full}^{(l-1),m} \right\|_F$$

$$\overset{(c)}{\le} C_\sigma B_W B_H^f B_{\Delta P}^f + C_\sigma B_W B_P \left\| \widetilde{\boldsymbol{H}}_{full}^{(l-1),m} - \boldsymbol{H}_{full}^{(l-1),m} \right\|_F$$

$$\le \left( C_\sigma B_W B_H^f B_{\Delta P}^f \right)^l + \left( C_\sigma B_W B_P \right)^l \left\| \widetilde{\boldsymbol{X}}_{full}^m - \boldsymbol{X}_{full}^m \right\|_F$$

$$\overset{(d)}{\le} \left( C_\sigma B_W B_H^f B_{\Delta P}^f \right)^l + \left( C_\sigma B_W B_P \right)^l B_{\Delta X}^f$$

$$\le \max_{1 \le l \le L} \left( \left( C_\sigma B_W B_H^f B_{\Delta P}^f \right)^l + \left( C_\sigma B_W B_P \right)^l B_{\Delta X}^f \right), \tag{10}$$

where (a) follows from Assumptions 5.2 and 5.3, (b) is due to Assumption 5.3 and Lemma D.3, and (c) and (d) are because of Proposition D.1.

$$\left\| \widetilde{\boldsymbol{D}}_{local}^{(l),m} - \boldsymbol{D}_{local}^{(l),m} \right\|_F$$

$$= \left\| \left[ \widetilde{\boldsymbol{P}}_{local}^{(l+1),m} \right]^\top \widetilde{\boldsymbol{D}}_{local}^{(l+1),m} \circ \nabla \sigma \left( \widetilde{\boldsymbol{Z}}_{local}^{(l+1),m} \right) \left[ \boldsymbol{W}^{(l+1),m} \right]^\top \right.$$

$$- \left[\boldsymbol{P}_{local}^{(l+1),m}\right]^{\top} \boldsymbol{D}_{local}^{(l+1),m} \circ \nabla\sigma\left(\boldsymbol{Z}_{local}^{(l+1),m}\right) \left[\boldsymbol{W}^{(l+1),m}\right]^{\top}\bigg\|_F$$

$$\overset{(a)}{\leq} B_W \left\|\left[\widetilde{\boldsymbol{P}}_{local}^{(l+1),m}\right]^{\top} \widetilde{\boldsymbol{D}}_{local}^{(l+1),m}\circ\nabla\sigma\left(\widetilde{\boldsymbol{Z}}_{local}^{(l+1),m}\right) - \left[\boldsymbol{P}_{local}^{(l+1),m}\right]^{\top} \boldsymbol{D}_{local}^{(l+1),m}\circ\nabla\sigma\left(\boldsymbol{Z}_{local}^{(l+1),m}\right)\right\|_F$$

$$\leq B_W \left\|\left[\widetilde{\boldsymbol{P}}_{local}^{(l+1),m}\right]^{\top} \widetilde{\boldsymbol{D}}_{local}^{(l+1),m}\circ\nabla\sigma\left(\widetilde{\boldsymbol{Z}}_{local}^{(l+1),m}\right) - \left[\boldsymbol{P}_{local}^{(l+1),m}\right]^{\top} \widetilde{\boldsymbol{D}}_{local}^{(l+1),m}\circ\nabla\sigma\left(\widetilde{\boldsymbol{Z}}_{local}^{(l+1),m}\right)\right\|_F$$

$$+ B_W \left\|\left[\boldsymbol{P}_{local}^{(l+1),m}\right]^{\top} \widetilde{\boldsymbol{D}}_{local}^{(l+1),m}\circ\nabla\sigma\left(\widetilde{\boldsymbol{Z}}_{local}^{(l+1),m}\right) - \left[\boldsymbol{P}_{local}^{(l+1),m}\right]^{\top} \boldsymbol{D}_{local}^{(l+1),m}\circ\nabla\sigma\left(\widetilde{\boldsymbol{Z}}_{local}^{(l+1),m}\right)\right\|_F$$

$$+ B_W \left\|\left[\boldsymbol{P}_{local}^{(l+1),m}\right]^{\top} \boldsymbol{D}_{local}^{(l+1),m}\circ\nabla\sigma\left(\widetilde{\boldsymbol{Z}}_{local}^{(l+1),m}\right) - \left[\boldsymbol{P}_{local}^{(l+1),m}\right]^{\top} \boldsymbol{D}_{local}^{(l+1),m}\circ\nabla\sigma\left(\boldsymbol{Z}_{local}^{(l+1),m}\right)\right\|_F$$

$$\overset{(b)}{\leq} B_W B_D^l C_\sigma \left\|\widetilde{\boldsymbol{P}}_{local}^{(l+1),m} - \boldsymbol{P}_{local}^{(l+1),m}\right\|_F + B_W B_P C_\sigma \left\|\widetilde{\boldsymbol{D}}_{local}^{(l+1),m} - \boldsymbol{D}_{local}^{(l+1),m}\right\|_F$$

$$+ B_W B_P B_D^l \left\|\nabla\sigma\left(\widetilde{\boldsymbol{Z}}_{local}^{(l+1),m}\right) - \nabla\sigma\left(\boldsymbol{Z}_{local}^{(l+1),m}\right)\right\|_F$$

$$\overset{(c)}{\leq} B_W B_D^l C_\sigma \left\|\widetilde{\boldsymbol{P}}_{local}^{(l+1),m} - \boldsymbol{P}_{local}^{(l+1),m}\right\|_F + B_W B_P C_\sigma \left\|\widetilde{\boldsymbol{D}}_{local}^{(l+1),m} - \boldsymbol{D}_{local}^{(l+1),m}\right\|_F$$

$$+ B_W^2 B_P B_D^l L_\sigma \left\|\widetilde{\boldsymbol{P}}_{local}^{(l),m} \widetilde{\boldsymbol{H}}_{local}^{(l-1),m} - \boldsymbol{P}_{local}^{(l),m} \boldsymbol{H}_{local}^{(l-1),m}\right\|_F$$

$$\leq B_W B_D^l C_\sigma \left\|\widetilde{\boldsymbol{P}}_{local}^{(l+1),m} - \boldsymbol{P}_{local}^{(l+1),m}\right\|_F + B_W B_P C_\sigma \left\|\widetilde{\boldsymbol{D}}_{local}^{(l+1),m} - \boldsymbol{D}_{local}^{(l+1),m}\right\|_F$$

$$+ B_W^2 B_P B_D^l L_\sigma \left\|\widetilde{\boldsymbol{P}}_{local}^{(l),m} \widetilde{\boldsymbol{H}}_{local}^{(l-1),m} - \boldsymbol{P}_{local}^{(l),m} \widetilde{\boldsymbol{H}}_{local}^{(l-1),m}\right\|_F$$

$$+ B_W^2 B_P B_D^l L_\sigma \left\|\boldsymbol{P}_{local}^{(l),m} \widetilde{\boldsymbol{H}}_{local}^{(l-1),m} - \boldsymbol{P}_{local}^{(l),m} \boldsymbol{H}_{local}^{(l-1),m}\right\|_F$$

$$\overset{(d)}{\leq} B_W B_D^l C_\sigma \left\|\widetilde{\boldsymbol{P}}_{local}^{(l+1),m} - \boldsymbol{P}_{local}^{(l+1),m}\right\|_F + B_W B_P C_\sigma \left\|\widetilde{\boldsymbol{D}}_{local}^{(l+1),m} - \boldsymbol{D}_{local}^{(l+1),m}\right\|_F$$

$$+ B_W^2 B_P B_D^l L_\sigma B_H^l \left\|\widetilde{\boldsymbol{P}}_{local}^{(l),m} - \boldsymbol{P}_{local}^{(l),m}\right\|_F + B_W^2 B_P^2 B_D^l L_\sigma \left\|\widetilde{\boldsymbol{H}}_{local}^{(l-1),m} - \boldsymbol{H}_{local}^{(l-1),m}\right\|_F$$

$$\overset{(e)}{\leq} B_W B_D^l C_\sigma B_{\Delta P}^l + B_W^2 B_P B_D^l L_\sigma B_H^l B_{\Delta P}^l + B_W^2 B_P^2 B_D^l L_\sigma B_{\Delta H}^l$$

$$+ B_W B_P C_\sigma \left\|\widetilde{\boldsymbol{D}}_{local}^{(l+1),m} - \boldsymbol{D}_{local}^{(l+1),m}\right\|_F$$

$$\leq \left(B_W B_D^l C_\sigma B_{\Delta P}^l + B_W^2 B_P B_D^l L_\sigma B_H^l B_{\Delta P}^l + B_W^2 B_P^2 B_D^l L_\sigma B_{\Delta H}^l\right)^{L-l}$$

$$+ \left(B_W B_P C_\sigma\right)^{L-l} \left\|\widetilde{\boldsymbol{D}}_{local}^{(L),m} - \boldsymbol{D}_{local}^{(L),m}\right\|_F$$

$$\overset{(f)}{\leq} \left(B_W B_D^l C_\sigma B_{\Delta P}^l + B_W^2 B_P B_D^l L_\sigma B_H^l B_{\Delta P}^l + B_W^2 B_P^2 B_D^l L_\sigma B_{\Delta H}^l\right)^{L-l}$$

$$+ \left(B_W B_P C_\sigma\right)^{L-l} L_l \left\|\widetilde{\boldsymbol{H}}_{local}^{(L),m} - \boldsymbol{H}_{local}^{(L),m}\right\|_F$$

$$\overset{(g)}{\leq} \left(B_W B_D^l C_\sigma B_{\Delta P}^l + B_W^2 B_P B_D^l L_\sigma B_H^l B_{\Delta P}^l + B_W^2 B_P^2 B_D^l L_\sigma B_{\Delta H}^l\right)^{L-l}$$

$$+ \left(B_W B_P C_\sigma\right)^{L-l} L_l B_{\Delta H}^l$$

$$\leq \max_{1\leq l\leq L} \left(\left(B_W B_D^l C_\sigma B_{\Delta P}^l + B_W^2 B_P B_D^l L_\sigma B_H^l B_{\Delta P}^l + B_W^2 B_P^2 B_D^l L_\sigma B_{\Delta H}^l\right)^{L-l}\right.$$

$$\left.+ \left(B_W B_P C_\sigma\right)^{L-l} L_l B_{\Delta H}^l\right),$$

where (a) uses Assumption 5.3, (b) is because of Assumptions 5.2 and 5.3 and Lemma D.3, (c) follows from Assumptions 5.2 and 5.3, (d) utilizes Assumption 5.3 and Lemma D.3, (e) results from Eq. (9) and Proposition D.1, (f) is because of Assumption 5.1, and (g) is due to Eq. (9).

$$\left\|\widetilde{\boldsymbol{D}}_{full}^{(l),m} - \boldsymbol{D}_{full}^{(l),m}\right\|_F$$

$$
= \left\| \left[ \widetilde{\boldsymbol{P}}_{local}^{(l+1),m} + \widetilde{\boldsymbol{P}}_{remote}^{(l+1),m} \right]^{\top} \widetilde{\boldsymbol{D}}_{full}^{(l+1),m} \circ \nabla \sigma \left( \widetilde{\boldsymbol{Z}}_{full}^{(l+1),m} \right) \left[ \boldsymbol{W}^{(l+1),m} \right]^{\top} \right.
$$

$$
\left. - \left[ \boldsymbol{P}_{full}^{(l+1),m} \right]^{\top} \boldsymbol{D}_{full}^{(l+1),m} \circ \nabla \sigma \left( \boldsymbol{Z}_{full}^{(l+1),m} \right) \left[ \boldsymbol{W}^{(l+1),m} \right]^{\top} \right\|_F
$$

$$
\overset{(a)}{\leq} B_W \left\| \left[ \widetilde{\boldsymbol{P}}_{full}^{(l+1),m} \right]^{\top} \widetilde{\boldsymbol{D}}_{full}^{(l+1),m} \circ \nabla \sigma \left( \widetilde{\boldsymbol{Z}}_{full}^{(l+1),m} \right) - \left[ \boldsymbol{P}_{full}^{(l+1),m} \right]^{\top} \boldsymbol{D}_{full}^{(l+1),m} \circ \nabla \sigma \left( \boldsymbol{Z}_{full}^{(l+1),m} \right) \right\|_F
$$

$$
\leq B_W \left\| \left[ \widetilde{\boldsymbol{P}}_{full}^{(l+1),m} \right]^{\top} \widetilde{\boldsymbol{D}}_{full}^{(l+1),m} \circ \nabla \sigma \left( \widetilde{\boldsymbol{Z}}_{full}^{(l+1),m} \right) - \left[ \boldsymbol{P}_{full}^{(l+1),m} \right]^{\top} \widetilde{\boldsymbol{D}}_{full}^{(l+1),m} \circ \nabla \sigma \left( \widetilde{\boldsymbol{Z}}_{full}^{(l+1),m} \right) \right\|_F
$$

$$
+ B_W \left\| \left[ \boldsymbol{P}_{full}^{(l+1),m} \right]^{\top} \widetilde{\boldsymbol{D}}_{full}^{(l+1),m} \circ \nabla \sigma \left( \widetilde{\boldsymbol{Z}}_{full}^{(l+1),m} \right) - \left[ \boldsymbol{P}_{full}^{(l+1),m} \right]^{\top} \boldsymbol{D}_{full}^{(l+1),m} \circ \nabla \sigma \left( \widetilde{\boldsymbol{Z}}_{full}^{(l+1),m} \right) \right\|_F
$$

$$
+ B_W \left\| \left[ \boldsymbol{P}_{full}^{(l+1),m} \right]^{\top} \boldsymbol{D}_{full}^{(l+1),m} \circ \nabla \sigma \left( \widetilde{\boldsymbol{Z}}_{full}^{(l+1),m} \right) - \left[ \boldsymbol{P}_{full}^{(l+1),m} \right]^{\top} \boldsymbol{D}_{full}^{(l+1),m} \circ \nabla \sigma \left( \boldsymbol{Z}_{full}^{(l+1),m} \right) \right\|_F
$$

$$
\overset{(b)}{\leq} B_W B_D^f C_\sigma \left\| \widetilde{\boldsymbol{P}}_{full}^{(l+1),m} - \boldsymbol{P}_{full}^{(l+1),m} \right\|_F + B_W B_P C_\sigma \left\| \widetilde{\boldsymbol{D}}_{full}^{(l+1),m} - \boldsymbol{D}_{full}^{(l+1),m} \right\|_F
$$

$$
+ B_W B_P B_D^f \left\| \nabla \sigma \left( \widetilde{\boldsymbol{Z}}_{full}^{(l+1),m} \right) - \nabla \sigma \left( \boldsymbol{Z}_{full}^{(l+1),m} \right) \right\|_F
$$

$$
\overset{(c)}{\leq} B_W B_D^f C_\sigma \left\| \widetilde{\boldsymbol{P}}_{full}^{(l+1),m} - \boldsymbol{P}_{full}^{(l+1),m} \right\|_F + B_W B_P C_\sigma \left\| \widetilde{\boldsymbol{D}}_{full}^{(l+1),m} - \boldsymbol{D}_{full}^{(l+1),m} \right\|_F
$$

$$
+ B_W^2 B_P B_D^f L_\sigma \left\| \widetilde{\boldsymbol{P}}_{full}^{(l+1),m} \widetilde{\boldsymbol{H}}_{full}^{(l),m} - \boldsymbol{P}_{full}^{(l+1),m} \boldsymbol{H}_{full}^{(l),m} \right\|_F
$$

$$
\leq B_W B_D^f C_\sigma \left\| \widetilde{\boldsymbol{P}}_{full}^{(l+1),m} - \boldsymbol{P}_{full}^{(l+1),m} \right\|_F + B_W B_P C_\sigma \left\| \widetilde{\boldsymbol{D}}_{full}^{(l+1),m} - \boldsymbol{D}_{full}^{(l+1),m} \right\|_F
$$

$$
+ B_W^2 B_P B_D^f L_\sigma \left\| \widetilde{\boldsymbol{P}}_{full}^{(l+1),m} \widetilde{\boldsymbol{H}}_{full}^{(l),m} - \boldsymbol{P}_{full}^{(l+1),m} \widetilde{\boldsymbol{H}}_{full}^{(l),m} \right\|_F
$$

$$
+ B_W^2 B_P B_D^f L_\sigma \left\| \boldsymbol{P}_{full}^{(l+1),m} \widetilde{\boldsymbol{H}}_{full}^{(l),m} - \boldsymbol{P}_{full}^{(l+1),m} \boldsymbol{H}_{full}^{(l),m} \right\|_F
$$

$$
\overset{(d)}{\leq} B_W B_D^f C_\sigma \left\| \widetilde{\boldsymbol{P}}_{full}^{(l+1),m} - \boldsymbol{P}_{full}^{(l+1),m} \right\|_F + B_W B_P C_\sigma \left\| \widetilde{\boldsymbol{D}}_{full}^{(l+1),m} - \boldsymbol{D}_{full}^{(l+1),m} \right\|_F
$$

$$
+ B_W^2 B_P B_D^f L_\sigma B_H^f \left\| \widetilde{\boldsymbol{P}}_{full}^{(l+1),m} - \boldsymbol{P}_{full}^{(l+1),m} \right\|_F + B_W^2 B_P^2 B_D^f L_\sigma \left\| \widetilde{\boldsymbol{H}}_{full}^{(l),m} - \boldsymbol{H}_{full}^{(l),m} \right\|_F
$$

$$
\overset{(e)}{\leq} B_W B_D^f C_\sigma B_{\Delta P}^f + B_W^2 B_P B_D^f L_\sigma B_H^f B_{\Delta P}^f + B_W^2 B_P^2 B_D^f L_\sigma B_{\Delta H}^f
$$

$$
+ B_W B_P C_\sigma \left\| \widetilde{\boldsymbol{D}}_{full}^{(l+1),m} - \boldsymbol{D}_{full}^{(l+1),m} \right\|_F
$$

$$
\leq \left( B_W B_D^f C_\sigma B_{\Delta P}^f + B_W^2 B_P B_D^f L_\sigma B_H^f B_{\Delta P}^f + B_W^2 B_P^2 B_D^f L_\sigma B_{\Delta H}^f \right)^{L-l}
$$

$$
+ \left( B_W B_P C_\sigma \right)^{L-l} \left\| \widetilde{\boldsymbol{D}}_{full}^{(L),m} - \boldsymbol{D}_{full}^{(L),m} \right\|_F
$$

$$
\overset{(f)}{\leq} \left( B_W B_D^f C_\sigma B_{\Delta P}^f + B_W^2 B_P B_D^f L_\sigma B_H^f B_{\Delta P}^f + B_W^2 B_P^2 B_D^f L_\sigma B_{\Delta H}^f \right)^{L-l}
$$

$$
+ \left( B_W B_P C_\sigma \right)^{L-l} L_l \left\| \widetilde{\boldsymbol{H}}_{full}^{(L),m} - \boldsymbol{H}_{full}^{(L),m} \right\|_F
$$

$$
\overset{(g)}{\leq} \left( B_W B_D^f C_\sigma B_{\Delta P}^f + B_W^2 B_P B_D^f L_\sigma B_H^f B_{\Delta P}^f + B_W^2 B_P^2 B_D^f L_\sigma B_{\Delta H}^f \right)^{L-l}
$$

$$
+ \left( B_W B_P C_\sigma \right)^{L-l} L_l B_{\Delta H}^f
$$

$$
\leq \max_{1 \leq l \leq L} \left( \left( B_W B_D^f C_\sigma B_{\Delta P}^f + B_W^2 B_P B_D^f L_\sigma B_H^f B_{\Delta P}^f + B_W^2 B_P^2 B_D^f L_\sigma B_{\Delta H}^f \right)^{L-l} \right.
$$

$$
\left. + \left( B_W B_P C_\sigma \right)^{L-l} L_l B_{\Delta H}^f \right),
$$

where (a) is because of Assumption 5.3, (b) results from Assumptions 5.2 and 5.3 and Lemma D.3, (c) uses Assumptions 5.2 and 5.3, (d) is due to Assumption 5.3 and Lemma D.3, (e) follows from Eq. (10) and Proposition D.1, (f) utilizes Assumption 5.1, and (g) is because of Eq. (10).

$\square$

**Lemma D.5.** *Under Assumptions 5.1–5.3, and for any $l \in [L]$, the errors caused by the information loss of the cross-client neighbors are bounded, i.e.,*

$$\left\| \boldsymbol{H}_{local}^{(l),m} - \boldsymbol{H}_{full}^{(l),m} \right\|_F \leq B_{\Delta H}^r, \qquad \left\| \boldsymbol{D}_{local}^{(l),m} - \boldsymbol{D}_{full}^{(l),m} \right\|_F \leq B_{\Delta D}^r,$$

*where*

$$B_{\Delta H}^r = \max_{1 \leq l \leq L} \left( (C_\sigma B_W B_P)^l B_X^r + \left( C_\sigma B_W B_H^f B_P \right)^l \right),$$

$$B_{\Delta D}^r = \max_{1 \leq l \leq L} \left( \left( B_W B_D^l C_\sigma B_P + B_W^2 B_P^2 B_D^f L_\sigma B_H^l + B_W^2 B_P^2 B_D^f L_\sigma B_{\Delta H}^r \right)^{L-l} \right.$$

$$\left. + (B_W B_P C_\sigma)^{L-l} L_l B_{\Delta H}^r \right).$$

*Proof.*

$$\left\| \boldsymbol{H}_{local}^{(l),m} - \boldsymbol{H}_{full}^{(l),m} \right\|_F = \left\| \sigma \left( \boldsymbol{P}_{local}^{(l),m} \boldsymbol{H}_{local}^{(l-1),m} \right) \boldsymbol{W}^{(l),m} - \sigma \left( \boldsymbol{P}_{full}^{(l),m} \boldsymbol{H}_{full}^{(l-1),m} \right) \boldsymbol{W}^{(l),m} \right\|_F$$

$$\overset{(a)}{\leq} C_\sigma B_W \left\| \boldsymbol{P}_{local}^{(l),m} \boldsymbol{H}_{local}^{(l-1),m} - \boldsymbol{P}_{full}^{(l),m} \boldsymbol{H}_{full}^{(l-1),m} \right\|_F$$

$$\leq C_\sigma B_W \left\| \boldsymbol{P}_{local}^{(l),m} \boldsymbol{H}_{local}^{(l-1),m} - \boldsymbol{P}_{local}^{(l),m} \boldsymbol{H}_{full}^{(l-1),m} \right\|_F$$

$$+ C_\sigma B_W \left\| \boldsymbol{P}_{local}^{(l),m} \boldsymbol{H}_{full}^{(l-1),m} - \boldsymbol{P}_{full}^{(l),m} \boldsymbol{H}_{full}^{(l-1),m} \right\|_F$$

$$\overset{(b)}{\leq} C_\sigma B_W B_P \left\| \boldsymbol{H}_{local}^{(l-1),m} - \boldsymbol{H}_{full}^{(l-1),m} \right\|_F + C_\sigma B_W B_H^f \left\| \boldsymbol{P}_{local}^{(l),m} - \boldsymbol{P}_{full}^{(l),m} \right\|_F$$

$$\leq C_\sigma B_W B_P \left\| \boldsymbol{H}_{local}^{(l-1),m} - \boldsymbol{H}_{full}^{(l-1),m} \right\|_F + C_\sigma B_W B_H^f \left\| \boldsymbol{P}_{remote}^{(l),m} \right\|_F$$

$$\overset{(c)}{\leq} C_\sigma B_W B_P \left\| \boldsymbol{H}_{local}^{(l-1),m} - \boldsymbol{H}_{full}^{(l-1),m} \right\|_F + C_\sigma B_W B_H^f B_P$$

$$\leq (C_\sigma B_W B_P)^l \left\| \boldsymbol{X}_{local}^m - \boldsymbol{X}_{full}^m \right\|_F + \left( C_\sigma B_W B_H^f B_P \right)^l$$

$$\overset{(d)}{\leq} (C_\sigma B_W B_P)^l B_X^r + \left( C_\sigma B_W B_H^f B_P \right)^l$$

$$\leq \max_{1 \leq l \leq L} \left( (C_\sigma B_W B_P)^l B_X^r + \left( C_\sigma B_W B_H^f B_P \right)^l \right), \tag{11}$$

where (a) uses Assumptions 5.2 and 5.3, (b) is because of Assumption 5.3 and Lemma D.3, (c) follows from Assumption 5.3, and (d) is due to Proposition D.1.

$$\left\| \boldsymbol{D}_{local}^{(l),m} - \boldsymbol{D}_{full}^{(l),m} \right\|_F$$

$$= \left\| \left[ \boldsymbol{P}_{local}^{(l+1),m} \right]^\top \boldsymbol{D}_{local}^{(l+1),m} \circ \nabla\sigma \left( \boldsymbol{Z}_{local}^{(l+1),m} \right) \left[ \boldsymbol{W}^{(l+1),m} \right]^\top \right.$$

$$- \left[ \boldsymbol{P}_{full}^{(l+1),m} \right]^\top \boldsymbol{D}_{full}^{(l+1),m} \circ \nabla\sigma \left( \boldsymbol{Z}_{full}^{(l+1),m} \right) \left[ \boldsymbol{W}^{(l+1),m} \right]^\top \Bigg\|_F$$

$$\overset{(a)}{\leq} B_W \left\| \left[ \boldsymbol{P}_{local}^{(l+1),m} \right]^\top \boldsymbol{D}_{local}^{(l+1),m} \circ \nabla\sigma \left( \boldsymbol{Z}_{local}^{(l+1),m} \right) - \left[ \boldsymbol{P}_{full}^{(l+1),m} \right]^\top \boldsymbol{D}_{full}^{(l+1),m} \circ \nabla\sigma \left( \boldsymbol{Z}_{full}^{(l+1),m} \right) \right\|_F$$

$$\leq B_W \left\| \left[ \boldsymbol{P}_{local}^{(l+1),m} \right]^\top \boldsymbol{D}_{local}^{(l+1),m} \circ \nabla\sigma \left( \boldsymbol{Z}_{local}^{(l+1),m} \right) - \left[ \boldsymbol{P}_{full}^{(l+1),m} \right]^\top \boldsymbol{D}_{local}^{(l+1),m} \circ \nabla\sigma \left( \boldsymbol{Z}_{local}^{(l+1),m} \right) \right\|_F$$

$$+ B_W \left\| \left[ \boldsymbol{P}_{full}^{(l+1),m} \right]^\top \boldsymbol{D}_{local}^{(l+1),m} \circ \nabla\sigma\left( \boldsymbol{Z}_{local}^{(l+1),m} \right) - \left[ \boldsymbol{P}_{full}^{(l+1),m} \right]^\top \boldsymbol{D}_{full}^{(l+1),m} \circ \nabla\sigma\left( \boldsymbol{Z}_{local}^{(l+1),m} \right) \right\|_F$$

$$+ B_W \left\| \left[ \boldsymbol{P}_{full}^{(l+1),m} \right]^\top \boldsymbol{D}_{full}^{(l+1),m} \circ \nabla\sigma\left( \boldsymbol{Z}_{local}^{(l+1),m} \right) - \left[ \boldsymbol{P}_{full}^{(l+1),m} \right]^\top \boldsymbol{D}_{full}^{(l+1),m} \circ \nabla\sigma\left( \boldsymbol{Z}_{full}^{(l+1),m} \right) \right\|_F$$

$$\overset{(b)}{\leq} B_W B_D^l C_\sigma \left\| \boldsymbol{P}_{local}^{(l+1),m} - \boldsymbol{P}_{full}^{(l+1),m} \right\|_F + B_W B_P C\sigma \left\| \boldsymbol{D}_{local}^{(l+1),m} - \boldsymbol{D}_{full}^{(l+1),m} \right\|_F$$

$$+ B_W B_P B_D^f \left\| \nabla\sigma\left( \boldsymbol{Z}_{local}^{(l+1),m} \right) - \nabla\sigma\left( \boldsymbol{Z}_{full}^{(l+1),m} \right) \right\|_F$$

$$\overset{(c)}{\leq} B_W B_D^l C_\sigma \left\| \boldsymbol{P}_{local}^{(l+1),m} - \boldsymbol{P}_{full}^{(l+1),m} \right\|_F + B_W B_P C\sigma \left\| \boldsymbol{D}_{local}^{(l+1),m} - \boldsymbol{D}_{full}^{(l+1),m} \right\|_F$$

$$+ B_W^2 B_P B_D^f L_\sigma \left\| \boldsymbol{P}_{local}^{(l+1),m} \boldsymbol{H}_{local}^{(l),m} - \boldsymbol{P}_{full}^{(l+1),m} \boldsymbol{H}_{full}^{(l),m} \right\|_F$$

$$\leq B_W B_D^l C_\sigma \left\| \boldsymbol{P}_{local}^{(l+1),m} - \boldsymbol{P}_{full}^{(l+1),m} \right\|_F + B_W B_P C\sigma \left\| \boldsymbol{D}_{local}^{(l+1),m} - \boldsymbol{D}_{full}^{(l+1),m} \right\|_F$$

$$+ B_W^2 B_P B_D^f L_\sigma \left\| \boldsymbol{P}_{local}^{(l+1),m} \boldsymbol{H}_{local}^{(l),m} - \boldsymbol{P}_{full}^{(l+1),m} \boldsymbol{H}_{local}^{(l),m} \right\|_F$$

$$+ B_W^2 B_P B_D^f L_\sigma \left\| \boldsymbol{P}_{full}^{(l+1),m} \boldsymbol{H}_{local}^{(l),m} - \boldsymbol{P}_{full}^{(l+1),m} \boldsymbol{H}_{full}^{(l),m} \right\|_F$$

$$\overset{(d)}{\leq} B_W B_D^l C_\sigma \left\| \boldsymbol{P}_{local}^{(l+1),m} - \boldsymbol{P}_{full}^{(l+1),m} \right\|_F + B_W B_P C\sigma \left\| \boldsymbol{D}_{local}^{(l+1),m} - \boldsymbol{D}_{full}^{(l+1),m} \right\|_F$$

$$+ B_W^2 B_P B_D^f L_\sigma B_H^l \left\| \boldsymbol{P}_{local}^{(l+1),m} - \boldsymbol{P}_{full}^{(l+1),m} \right\|_F + B_W^2 B_P^2 B_D^f L_\sigma \left\| \boldsymbol{H}_{local}^{(l),m} - \boldsymbol{H}_{full}^{(l),m} \right\|_F$$

$$\overset{(e)}{\leq} B_W B_D^l C_\sigma B_P + B_W^2 B_P^2 B_D^f L_\sigma B_H^l + B_W^2 B_P^2 B_D^f L_\sigma B_{\Delta H}^r$$

$$+ B_W B_P C\sigma \left\| \boldsymbol{D}_{local}^{(l+1),m} - \boldsymbol{D}_{full}^{(l+1),m} \right\|_F$$

$$\leq \left( B_W B_D^l C_\sigma B_P + B_W^2 B_P^2 B_D^f L_\sigma B_H^l + B_W^2 B_P^2 B_D^f L_\sigma B_{\Delta H}^r \right)^{L-l}$$

$$+ \left( B_W B_P C\sigma \right)^{L-l} \left\| \boldsymbol{D}_{local}^{(L),m} - \boldsymbol{D}_{full}^{(L),m} \right\|_F$$

$$\overset{(f)}{\leq} \left( B_W B_D^l C_\sigma B_P + B_W^2 B_P^2 B_D^f L_\sigma B_H^l + B_W^2 B_P^2 B_D^f L_\sigma B_{\Delta H}^r \right)^{L-l}$$

$$+ \left( B_W B_P C\sigma \right)^{L-l} L_l \left\| \boldsymbol{H}_{local}^{(L),m} - \boldsymbol{H}_{full}^{(L),m} \right\|_F$$

$$\overset{(g)}{\leq} \left( B_W B_D^l C_\sigma B_P + B_W^2 B_P^2 B_D^f L_\sigma B_H^l + B_W^2 B_P^2 B_D^f L_\sigma B_{\Delta H}^r \right)^{L-l}$$

$$+ \left( B_W B_P C\sigma \right)^{L-l} L_l B_{\Delta H}^r$$

$$\leq \max_{1 \leq l \leq L} \left( \left( B_W B_D^l C_\sigma B_P + B_W^2 B_P^2 B_D^f L_\sigma B_H^l + B_W^2 B_P^2 B_D^f L_\sigma B_{\Delta H}^r \right)^{L-l} \right.$$

$$\left. + \left( B_W B_P C\sigma \right)^{L-l} L_l B_{\Delta H}^r \right),$$

where (a) follows from Assumption 5.3, (b) uses Assumptions 5.2 and 5.3 and Lemma D.3, (c) is because of Assumptions 5.2 and 5.3, (d) results from Assumption 5.3 and Lemma D.3, (e) is due to Assumption 5.3 and Eq. (11), (f) utilizes Assumption 5.1, and (g) uses Eq. (11).

$\square$

## D.3 ERRORS OF STOCHASTIC GRADIENTS

**Lemma D.6.** *Under Assumptions 5.1–5.3, the errors between the stochastic gradients and the full gradients are bounded as follows:*

$$\left\| \nabla F_{local}^m \left( \boldsymbol{\theta}^m \right) - \nabla \widetilde{F}_{local}^m \left( \boldsymbol{\theta}^m \right) \right\|_F \leq L B_{\Delta G}^l, \quad \left\| \nabla F_{full}^m \left( \boldsymbol{\theta}^m \right) - \nabla \widetilde{F}_{full}^m \left( \boldsymbol{\theta}^m \right) \right\|_F \leq L B_{\Delta G}^f,$$

*where*

$$B_{\Delta G}^l = \max_{1 \le l \le L} \Big( \big(B_D^l C_\sigma + B_P B_H^l B_D^l L_\sigma B_W\big) B_H^l B_{\Delta P}^l + B_P B_H^l C_\sigma B_{\Delta D}^l$$

$$+ \big(B_D^l C_\sigma + B_P B_H^l B_D^l L_\sigma B_W\big) B_P B_{\Delta H}^l \Big), \tag{12}$$

$$B_{\Delta G}^f = \max_{1 \le l \le L} \Big( \big(B_D^f C_\sigma + B_P B_H^f B_D^f L_\sigma B_W\big) B_H^f B_{\Delta P}^f + B_P B_H^f C_\sigma B_{\Delta D}^f$$

$$+ \big(B_D^f C_\sigma + B_P B_H^f B_D^f L_\sigma B_W\big) B_P B_{\Delta H}^f \Big) \tag{13}$$

*Proof.*

$$\left\| \widetilde{\boldsymbol{G}}_{local}^{(l),m} - \boldsymbol{G}_{local}^{(l),m} \right\|_F$$

$$= \left\| \left[ \widetilde{\boldsymbol{P}}_{local}^{(l),m} \widetilde{\boldsymbol{H}}_{local}^{(l-1),m} \right]^\top \widetilde{\boldsymbol{D}}_{local}^{(l),m} \circ \nabla\sigma\left(\widetilde{\boldsymbol{Z}}_{local}^{(l),m}\right) - \left[ \boldsymbol{P}_{local}^{(l),m} \boldsymbol{H}_{local}^{(l-1),m} \right]^\top \boldsymbol{D}_{local}^{(l),m} \circ \nabla\sigma\left(\boldsymbol{Z}_{local}^{(l),m}\right) \right\|_F$$

$$\le \left\| \left[ \widetilde{\boldsymbol{P}}_{local}^{(l),m} \widetilde{\boldsymbol{H}}_{local}^{(l-1),m} \right]^\top \widetilde{\boldsymbol{D}}_{local}^{(l),m} \circ \nabla\sigma\left(\widetilde{\boldsymbol{Z}}_{local}^{(l),m}\right) - \left[ \boldsymbol{P}_{local}^{(l),m} \boldsymbol{H}_{local}^{(l-1),m} \right]^\top \widetilde{\boldsymbol{D}}_{local}^{(l),m} \circ \nabla\sigma\left(\widetilde{\boldsymbol{Z}}_{local}^{(l),m}\right) \right\|_F$$

$$+ \left\| \left[ \boldsymbol{P}_{local}^{(l),m} \boldsymbol{H}_{local}^{(l-1),m} \right]^\top \widetilde{\boldsymbol{D}}_{local}^{(l),m} \circ \nabla\sigma\left(\widetilde{\boldsymbol{Z}}_{local}^{(l),m}\right) - \left[ \boldsymbol{P}_{local}^{(l),m} \boldsymbol{H}_{local}^{(l-1),m} \right]^\top \boldsymbol{D}_{local}^{(l),m} \circ \nabla\sigma\left(\widetilde{\boldsymbol{Z}}_{local}^{(l),m}\right) \right\|_F$$

$$+ \left\| \left[ \boldsymbol{P}_{local}^{(l),m} \boldsymbol{H}_{local}^{(l-1),m} \right]^\top \boldsymbol{D}_{local}^{(l),m} \circ \nabla\sigma\left(\widetilde{\boldsymbol{Z}}_{local}^{(l),m}\right) - \left[ \boldsymbol{P}_{local}^{(l),m} \boldsymbol{H}_{local}^{(l-1),m} \right]^\top \boldsymbol{D}_{local}^{(l),m} \circ \nabla\sigma\left(\boldsymbol{Z}_{local}^{(l),m}\right) \right\|_F$$

$$\overset{(a)}{\le} B_D^l C_\sigma \left\| \widetilde{\boldsymbol{P}}_{local}^{(l),m} \widetilde{\boldsymbol{H}}_{local}^{(l-1),m} - \boldsymbol{P}_{local}^{(l),m} \boldsymbol{H}_{local}^{(l-1),m} \right\|_F + B_P B_H^l C_\sigma \left\| \widetilde{\boldsymbol{D}}_{local}^{(l),m} - \boldsymbol{D}_{local}^{(l),m} \right\|_F$$

$$+ B_P B_H^l B_D^l \left\| \nabla\sigma\left(\widetilde{\boldsymbol{Z}}_{local}^{(l),m}\right) - \nabla\sigma\left(\boldsymbol{Z}_{local}^{(l),m}\right) \right\|_F$$

$$\overset{(b)}{\le} B_D^l C_\sigma \left\| \widetilde{\boldsymbol{P}}_{local}^{(l),m} \widetilde{\boldsymbol{H}}_{local}^{(l-1),m} - \boldsymbol{P}_{local}^{(l),m} \boldsymbol{H}_{local}^{(l-1),m} \right\|_F + B_P B_H^l C_\sigma \left\| \widetilde{\boldsymbol{D}}_{local}^{(l),m} - \boldsymbol{D}_{local}^{(l),m} \right\|_F$$

$$+ B_P B_H^l B_D^l L_\sigma B_W \left\| \widetilde{\boldsymbol{P}}_{local}^{(l),m} \widetilde{\boldsymbol{H}}_{local}^{(l-1),m} - \boldsymbol{P}_{local}^{(l),m} \boldsymbol{H}_{local}^{(l-1),m} \right\|_F$$

$$\le \big(B_D^l C_\sigma + B_P B_H^l B_D^l L_\sigma B_W\big) \left\| \widetilde{\boldsymbol{P}}_{local}^{(l),m} \widetilde{\boldsymbol{H}}_{local}^{(l-1),m} - \boldsymbol{P}_{local}^{(l),m} \widetilde{\boldsymbol{H}}_{local}^{(l-1),m} \right\|_F$$

$$+ \big(B_D^l C_\sigma + B_P B_H^l B_D^l L_\sigma B_W\big) \left\| \boldsymbol{P}_{local}^{(l),m} \widetilde{\boldsymbol{H}}_{local}^{(l-1),m} - \boldsymbol{P}_{local}^{(l),m} \boldsymbol{H}_{local}^{(l-1),m} \right\|_F$$

$$+ B_P B_H^l C_\sigma \left\| \widetilde{\boldsymbol{D}}_{local}^{(l),m} - \boldsymbol{D}_{local}^{(l),m} \right\|_F$$

$$\overset{(c)}{\le} \big(B_D^l C_\sigma + B_P B_H^l B_D^l L_\sigma B_W\big) B_H^l \left\| \widetilde{\boldsymbol{P}}_{local}^{(l),m} - \boldsymbol{P}_{local}^{(l),m} \right\|_F + B_P B_H^l C_\sigma \left\| \widetilde{\boldsymbol{D}}_{local}^{(l),m} - \boldsymbol{D}_{local}^{(l),m} \right\|_F$$

$$+ \big(B_D^l C_\sigma + B_P B_H^l B_D^l L_\sigma B_W\big) B_P \left\| \widetilde{\boldsymbol{H}}_{local}^{(l-1),m} - \boldsymbol{H}_{local}^{(l-1),m} \right\|_F$$

$$\overset{(d)}{\le} \big(B_D^l C_\sigma + B_P B_H^l B_D^l L_\sigma B_W\big) B_H^l B_{\Delta P}^l + B_P B_H^l C_\sigma B_{\Delta D}^l$$

$$+ \big(B_D^l C_\sigma + B_P B_H^l B_D^l L_\sigma B_W\big) B_P B_{\Delta H}^l$$

$$\le \max_{1 \le l \le L} \Big( \big(B_D^l C_\sigma + B_P B_H^l B_D^l L_\sigma B_W\big) B_H^l B_{\Delta P}^l + B_P B_H^l C_\sigma B_{\Delta D}^l$$

$$+ \big(B_D^l C_\sigma + B_P B_H^l B_D^l L_\sigma B_W\big) B_P B_{\Delta H}^l \Big) := B_{\Delta G}^l,$$

where (a) follows from Assumptions 5.2 and 5.3 and Lemma D.3, (b) is because of Assumptions 5.2 and 5.3, (c) uses Assumption 5.3 and Lemma D.3, and (d) results from Lemma D.4 and Proposition D.1.

When client $m$ performs local training with only its local data, the error between the stochastic gradient and the full-gradient can be bounded as:

$$\left\| \nabla F_{local}^m \left( \boldsymbol{\theta}^m \right) - \nabla \widetilde{F}_{local}^m \left( \boldsymbol{\theta}^m \right) \right\|_F = \sum_{l=1}^{L} \left\| \boldsymbol{G}_{local}^{(l),m} - \widetilde{\boldsymbol{G}}_{local}^{(l),m} \right\|_F \leq L B_{\Delta G}^l.$$

$$\left\| \widetilde{\boldsymbol{G}}_{full}^{(l),m} - \boldsymbol{G}_{full}^{(l),m} \right\|_F$$

$$= \left\| \left[ \widetilde{\boldsymbol{P}}_{local}^{(l),m} \widetilde{\boldsymbol{H}}_{local}^{(l-1),m} + \widetilde{\boldsymbol{P}}_{remote}^{(l),m} \widetilde{\boldsymbol{H}}_{remote}^{(l-1),m} \right]^\top \widetilde{\boldsymbol{D}}_{full}^{(l),m} \circ \nabla \sigma \left( \widetilde{\boldsymbol{Z}}_{full}^{(l),m} \right) \right.$$

$$\left. - \left[ \boldsymbol{P}_{full}^{(l),m} \boldsymbol{H}_{full}^{(l-1),m} \right]^\top \boldsymbol{D}_{full}^{(l),m} \circ \nabla \sigma \left( \boldsymbol{Z}_{full}^{(l),m} \right) \right\|_F$$

$$\leq \left\| \left[ \widetilde{\boldsymbol{P}}_{full}^{(l),m} \widetilde{\boldsymbol{H}}_{full}^{(l-1),m} \right]^\top \widetilde{\boldsymbol{D}}_{full}^{(l),m} \circ \nabla \sigma \left( \widetilde{\boldsymbol{Z}}_{full}^{(l),m} \right) - \left[ \boldsymbol{P}_{full}^{(l),m} \boldsymbol{H}_{full}^{(l-1),m} \right]^\top \widetilde{\boldsymbol{D}}_{full}^{(l),m} \circ \nabla \sigma \left( \widetilde{\boldsymbol{Z}}_{full}^{(l),m} \right) \right\|_F$$

$$+ \left\| \left[ \boldsymbol{P}_{full}^{(l),m} \boldsymbol{H}_{full}^{(l-1),m} \right]^\top \widetilde{\boldsymbol{D}}_{full}^{(l),m} \circ \nabla \sigma \left( \widetilde{\boldsymbol{Z}}_{full}^{(l),m} \right) - \left[ \boldsymbol{P}_{full}^{(l),m} \boldsymbol{H}_{full}^{(l-1),m} \right]^\top \boldsymbol{D}_{full}^{(l),m} \circ \nabla \sigma \left( \widetilde{\boldsymbol{Z}}_{full}^{(l),m} \right) \right\|_F$$

$$+ \left\| \left[ \boldsymbol{P}_{full}^{(l),m} \boldsymbol{H}_{full}^{(l-1),m} \right]^\top \boldsymbol{D}_{full}^{(l),m} \circ \nabla \sigma \left( \widetilde{\boldsymbol{Z}}_{full}^{(l),m} \right) - \left[ \boldsymbol{P}_{full}^{(l),m} \boldsymbol{H}_{full}^{(l-1),m} \right]^\top \boldsymbol{D}_{full}^{(l),m} \circ \nabla \sigma \left( \boldsymbol{Z}_{full}^{(l),m} \right) \right\|_F$$

$$\overset{(a)}{\leq} B_D^f C_\sigma \left\| \widetilde{\boldsymbol{P}}_{full}^{(l),m} \widetilde{\boldsymbol{H}}_{full}^{(l-1),m} - \boldsymbol{P}_{full}^{(l),m} \boldsymbol{H}_{full}^{(l-1),m} \right\|_F + B_P B_H^f C_\sigma \left\| \widetilde{\boldsymbol{D}}_{full}^{(l),m} - \boldsymbol{D}_{full}^{(l),m} \right\|_F$$

$$+ B_P B_H^f B_D^f \left\| \nabla \sigma \left( \widetilde{\boldsymbol{Z}}_{full}^{(l),m} \right) - \nabla \sigma \left( \boldsymbol{Z}_{full}^{(l),m} \right) \right\|_F$$

$$\overset{(b)}{\leq} B_D^f C_\sigma \left\| \widetilde{\boldsymbol{P}}_{full}^{(l),m} \widetilde{\boldsymbol{H}}_{full}^{(l-1),m} - \boldsymbol{P}_{full}^{(l),m} \boldsymbol{H}_{full}^{(l-1),m} \right\|_F + B_P B_H^f C_\sigma \left\| \widetilde{\boldsymbol{D}}_{full}^{(l),m} - \boldsymbol{D}_{full}^{(l),m} \right\|_F$$

$$+ B_P B_H^f B_D^f L_\sigma B_W \left\| \widetilde{\boldsymbol{P}}_{full}^{(l),m} \widetilde{\boldsymbol{H}}_{full}^{(l-1),m} - \boldsymbol{P}_{full}^{(l),m} \boldsymbol{H}_{full}^{(l-1),m} \right\|_F$$

$$\leq \left( B_D^f C_\sigma + B_P B_H^f B_D^f L_\sigma B_W \right) \left\| \widetilde{\boldsymbol{P}}_{full}^{(l),m} \widetilde{\boldsymbol{H}}_{full}^{(l-1),m} - \boldsymbol{P}_{full}^{(l),m} \widetilde{\boldsymbol{H}}_{full}^{(l-1),m} \right\|_F$$

$$+ \left( B_D^f C_\sigma + B_P B_H^f B_D^f L_\sigma B_W \right) \left\| \boldsymbol{P}_{full}^{(l),m} \widetilde{\boldsymbol{H}}_{full}^{(l-1),m} - \boldsymbol{P}_{full}^{(l),m} \boldsymbol{H}_{full}^{(l-1),m} \right\|_F$$

$$+ B_P B_H^f C_\sigma \left\| \widetilde{\boldsymbol{D}}_{full}^{(l),m} - \boldsymbol{D}_{full}^{(l),m} \right\|_F$$

$$\overset{(c)}{\leq} \left( B_D^f C_\sigma + B_P B_H^f B_D^f L_\sigma B_W \right) B_H^f \left\| \widetilde{\boldsymbol{P}}_{full}^{(l),m} - \boldsymbol{P}_{full}^{(l),m} \right\|_F + B_P B_H^f C_\sigma \left\| \widetilde{\boldsymbol{D}}_{full}^{(l),m} - \boldsymbol{D}_{full}^{(l),m} \right\|_F$$

$$+ \left( B_D^f C_\sigma + B_P B_H^f B_D^f L_\sigma B_W \right) B_P \left\| \widetilde{\boldsymbol{H}}_{full}^{(l-1),m} - \boldsymbol{H}_{full}^{(l-1),m} \right\|_F$$

$$\overset{(d)}{\leq} \left( B_D^f C_\sigma + B_P B_H^f B_D^f L_\sigma B_W \right) B_H^f B_{\Delta P}^f + B_P B_H^f C_\sigma B_{\Delta D}^f$$

$$+ \left( B_D^f C_\sigma + B_P B_H^f B_D^f L_\sigma B_W \right) B_P B_{\Delta H}^f$$

$$\leq \max_{1 \leq l \leq L} \left( \left( B_D^f C_\sigma + B_P B_H^f B_D^f L_\sigma B_W \right) B_H^f B_{\Delta P}^f + B_P B_H^f C_\sigma B_{\Delta D}^f \right.$$

$$\left. + \left( B_D^f C_\sigma + B_P B_H^f B_D^f L_\sigma B_W \right) B_P B_{\Delta H}^f \right) := B_{\Delta G}^f,$$

where (a) results from Assumptions 5.2 and 5.3 and Lemma D.3, (b) uses Assumptions 5.2 and 5.3, (c) is due to Assumption 5.3 and Lemma D.3, and (d) is because of Lemma D.4 and Proposition D.1.

When client $m$ conducts cross-client training using its local data and the cross-client neighbors, the error between the stochastic gradient and the full-gradient can be bounded as:

$$\left\| \nabla F_{full}^m \left( \boldsymbol{\theta}^m \right) - \nabla \widetilde{F}_{full}^m \left( \boldsymbol{\theta}^m \right) \right\|_F = \sum_{l=1}^{L} \left\| \boldsymbol{G}_{full}^{(l),m} - \widetilde{\boldsymbol{G}}_{full}^{(l),m} \right\|_F \leq L B_{\Delta G}^f.$$

$\square$

**Lemma D.7.** *Under Assumptions 5.1–5.3, the error between the full gradient computed with both the local graph data and the cross-client neighbors and the full gradient computed with only the local graph data is upper-bounded as follows:*

$$\left\| \nabla F_{full}^m \left( \boldsymbol{\theta}^m \right) - \nabla F_{local}^m \left( \boldsymbol{\theta}^m \right) \right\|_F \leq L B_{\Delta G}^r,$$

*where*

$$
\begin{aligned}
B_{\Delta G}^r = \max_{1 \leq l \leq L} \Bigg( & \left( B_D^l C_\sigma + B_P B_H^f B_D^f L_\sigma B_W \right) B_P B_{\Delta H}^r + B_P B_H^f C_\sigma B_{\Delta D}^r \\
& + \left( B_D^l C_\sigma + B_P B_H^f B_D^f L_\sigma B_W \right) B_H^f B_P \Bigg)
\end{aligned}
\tag{14}
$$

*Proof.*

$$\left\| \boldsymbol{G}_{local}^{(l),m} - \boldsymbol{G}_{full}^{(l),m} \right\|_F$$

$$= \left\| \left[ \boldsymbol{P}_{local}^{(l),m} \boldsymbol{H}_{local}^{(l-1),m} \right]^\top \boldsymbol{D}_{local}^{(l),m} \circ \nabla\sigma \left( \boldsymbol{Z}_{local}^{(l),m} \right) - \left[ \boldsymbol{P}_{full}^{(l),m} \boldsymbol{H}_{full}^{(l-1),m} \right]^\top \boldsymbol{D}_{full}^{(l),m} \circ \nabla\sigma \left( \boldsymbol{Z}_{full}^{(l),m} \right) \right\|_F$$

$$\leq \left\| \left[ \boldsymbol{P}_{local}^{(l),m} \boldsymbol{H}_{local}^{(l-1),m} \right]^\top \boldsymbol{D}_{local}^{(l),m} \circ \nabla\sigma \left( \boldsymbol{Z}_{local}^{(l),m} \right) - \left[ \boldsymbol{P}_{full}^{(l),m} \boldsymbol{H}_{full}^{(l-1),m} \right]^\top \boldsymbol{D}_{local}^{(l),m} \circ \nabla\sigma \left( \boldsymbol{Z}_{local}^{(l),m} \right) \right\|_F$$

$$+ \left\| \left[ \boldsymbol{P}_{full}^{(l),m} \boldsymbol{H}_{full}^{(l-1),m} \right]^\top \boldsymbol{D}_{local}^{(l),m} \circ \nabla\sigma \left( \boldsymbol{Z}_{local}^{(l),m} \right) - \left[ \boldsymbol{P}_{full}^{(l),m} \boldsymbol{H}_{full}^{(l-1),m} \right]^\top \boldsymbol{D}_{full}^{(l),m} \circ \nabla\sigma \left( \boldsymbol{Z}_{local}^{(l),m} \right) \right\|_F$$

$$+ \left\| \left[ \boldsymbol{P}_{full}^{(l),m} \boldsymbol{H}_{full}^{(l-1),m} \right]^\top \boldsymbol{D}_{full}^{(l),m} \circ \nabla\sigma \left( \boldsymbol{Z}_{local}^{(l),m} \right) - \left[ \boldsymbol{P}_{full}^{(l),m} \boldsymbol{H}_{full}^{(l-1),m} \right]^\top \boldsymbol{D}_{full}^{(l),m} \circ \nabla\sigma \left( \boldsymbol{Z}_{full}^{(l),m} \right) \right\|_F$$

$$\overset{(a)}{\leq} B_D^l C_\sigma \left\| \boldsymbol{P}_{local}^{(l),m} \boldsymbol{H}_{local}^{(l-1),m} - \boldsymbol{P}_{full}^{(l),m} \boldsymbol{H}_{full}^{(l-1),m} \right\|_F + B_P B_H^f C_\sigma \left\| \boldsymbol{D}_{local}^{(l),m} - \boldsymbol{D}_{full}^{(l),m} \right\|_F$$

$$+ B_P B_H^f B_D^f \left\| \nabla\sigma \left( \boldsymbol{Z}_{local}^{(l),m} \right) - \nabla\sigma \left( \boldsymbol{Z}_{full}^{(l),m} \right) \right\|_F$$

$$\overset{(b)}{\leq} B_D^l C_\sigma \left\| \boldsymbol{P}_{local}^{(l),m} \boldsymbol{H}_{local}^{(l-1),m} - \boldsymbol{P}_{full}^{(l),m} \boldsymbol{H}_{full}^{(l-1),m} \right\|_F + B_P B_H^f C_\sigma \left\| \boldsymbol{D}_{local}^{(l),m} - \boldsymbol{D}_{full}^{(l),m} \right\|_F$$

$$+ B_P B_H^f B_D^f L_\sigma B_W \left\| \boldsymbol{P}_{local}^{(l),m} \boldsymbol{H}_{local}^{(l-1),m} - \boldsymbol{P}_{full}^{(l),m} \boldsymbol{H}_{full}^{(l-1),m} \right\|_F$$

$$\leq \left( B_D^l C_\sigma + B_P B_H^f B_D^f L_\sigma B_W \right) \left\| \boldsymbol{P}_{local}^{(l),m} \boldsymbol{H}_{local}^{(l-1),m} - \boldsymbol{P}_{local}^{(l),m} \boldsymbol{H}_{full}^{(l-1),m} \right\|_F$$

$$+ \left( B_D^l C_\sigma + B_P B_H^f B_D^f L_\sigma B_W \right) \left\| \boldsymbol{P}_{local}^{(l),m} \boldsymbol{H}_{full}^{(l-1),m} - \boldsymbol{P}_{full}^{(l),m} \boldsymbol{H}_{full}^{(l-1),m} \right\|_F$$

$$+ B_P B_H^f C_\sigma \left\| \boldsymbol{D}_{local}^{(l),m} - \boldsymbol{D}_{full}^{(l),m} \right\|_F$$

$$\overset{(c)}{\leq} \left( B_D^l C_\sigma + B_P B_H^f B_D^f L_\sigma B_W \right) B_P \left\| \boldsymbol{H}_{local}^{(l-1),m} - \boldsymbol{H}_{full}^{(l-1),m} \right\|_F$$

$$+ \left( B_D^l C_\sigma + B_P B_H^f B_D^f L_\sigma B_W \right) B_H^f \left\| \boldsymbol{P}_{local}^{(l),m} - \boldsymbol{P}_{full}^{(l),m} \right\|_F + B_P B_H^f C_\sigma \left\| \boldsymbol{D}_{local}^{(l),m} - \boldsymbol{D}_{full}^{(l),m} \right\|_F$$

$$\overset{(d)}{\leq} \left( B_D^l C_\sigma + B_P B_H^f B_D^f L_\sigma B_W \right) B_P B_{\Delta H}^r + B_P B_H^f C_\sigma B_{\Delta D}^r$$

$$+ \left( B_D^l C_\sigma + B_P B_H^f B_D^f L_\sigma B_W \right) B_H^f B_P$$

$$\leq \max_{1 \leq l \leq L} \Bigg( \left( B_D^l C_\sigma + B_P B_H^f B_D^f L_\sigma B_W \right) B_P B_{\Delta H}^r + B_P B_H^f C_\sigma B_{\Delta D}^r$$

$$+ \left( B_D^l C_\sigma + B_P B_H^f B_D^f L_\sigma B_W \right) B_H^f B_P \Bigg) = B_{\Delta G}^r,$$

where (a) is because of Assumptions 5.2 and 5.3 and Lemma D.3, (b) uses Assumptions 5.2 and 5.3, (c) follow from Assumption 5.3 and Lemma D.3, and (d) results from Assumption 5.3 and Lemma D.5.

The error between the full gradient computed with both the local graph data and the cross-client neighbors and the full gradient computed with only the local graph data is bounded as follows:

$$\left\| \nabla F_{full}^m \left( \boldsymbol{\theta}^m \right) - \nabla F_{local}^m \left( \boldsymbol{\theta}^m \right) \right\|_F = \sum_{l=1}^{L} \left\| \boldsymbol{G}_{local}^{(l),m} - \boldsymbol{G}_{full}^{(l),m} \right\|_F \le L B_{\Delta G}^r.$$

$\square$

### D.4 MAIN PROOF OF THEOREM 5.6

**Theorem D.8.** *Under Assumptions 5.1–5.3, choose step-size $\alpha = \min\left\{ \sqrt{M}/\sqrt{T}, 1/L_F \right\}$, where $L_F$ is the smoothness constant given in Lemma D.2. The output of* Swift-FedGNN *with a L-layer GNN satisfies:*

$$\frac{1}{T} \sum_{t=0}^{T-1} \|\nabla \mathcal{L} \left( \boldsymbol{\theta}_t \right)\|^2 \le \frac{2}{\sqrt{MT}} \left( \mathcal{L} \left( \boldsymbol{\theta}_0 \right) - \mathcal{L} \left( \boldsymbol{\theta}^* \right) \right) + \left( 1 - \frac{K}{IM} \right) L^2 \left( B_{\Delta G}^l + B_{\Delta G}^r \right)^2 + \frac{K}{IM} L^2 (B_{\Delta G}^f)^2.$$

*Proof.*

$\mathcal{L} \left( \boldsymbol{\theta}_{t+1} \right) - \mathcal{L} \left( \boldsymbol{\theta}_t \right)$

$\overset{(a)}{\le} \left\langle \nabla \mathcal{L} \left( \boldsymbol{\theta}_t \right), \boldsymbol{\theta}_{t+1} - \boldsymbol{\theta}_t \right\rangle + \frac{L_F}{2} \|\boldsymbol{\theta}_{t+1} - \boldsymbol{\theta}_t\|^2$

$\overset{(b)}{=} -\alpha \left\langle \nabla \mathcal{L} \left( \boldsymbol{\theta}_t \right), \frac{1}{M} \sum_{m \in \mathcal{M}} \nabla \widetilde{F}^m \left( \boldsymbol{\theta}_t^m \right) \right\rangle + \frac{L_F}{2} \alpha^2 \left\| \frac{1}{M} \sum_{m \in \mathcal{M}} \nabla \widetilde{F}^m \left( \boldsymbol{\theta}_t^m \right) \right\|^2$

$\overset{(c)}{=} -\frac{\alpha}{2} \left\| \nabla \mathcal{L} \left( \boldsymbol{\theta}_t \right) \right\|^2 - \frac{\alpha}{2} \left\| \frac{1}{M} \sum_{m \in \mathcal{M}} \nabla \widetilde{F}^m \left( \boldsymbol{\theta}_t^m \right) \right\|^2 + \frac{\alpha}{2} \left\| \nabla \mathcal{L} \left( \boldsymbol{\theta}_t \right) - \frac{1}{M} \sum_{m \in \mathcal{M}} \nabla \widetilde{F}^m \left( \boldsymbol{\theta}_t^m \right) \right\|^2$

$+ \frac{L_F}{2} \alpha^2 \left\| \frac{1}{M} \sum_{m \in \mathcal{M}} \nabla \widetilde{F}^m \left( \boldsymbol{\theta}_t^m \right) \right\|^2$

$= -\frac{\alpha}{2} \left\| \nabla \mathcal{L} \left( \boldsymbol{\theta}_t \right) \right\|^2 - \frac{\alpha}{2} \left\| \frac{1}{M} \sum_{m \in \mathcal{M}} \nabla \widetilde{F}^m \left( \boldsymbol{\theta}_t^m \right) \right\|^2 + \frac{\alpha}{2} \left\| \frac{1}{M} \sum_{m \in \mathcal{M}} \left( \nabla F^m \left( \boldsymbol{\theta}_t^m \right) - \nabla \widetilde{F}^m \left( \boldsymbol{\theta}_t^m \right) \right) \right\|^2$

$+ \frac{L_F}{2} \alpha^2 \left\| \frac{1}{M} \sum_{m \in \mathcal{M}} \nabla \widetilde{F}^m \left( \boldsymbol{\theta}_t^m \right) \right\|^2$

$\overset{(d)}{\le} -\frac{\alpha}{2} \left\| \nabla \mathcal{L} \left( \boldsymbol{\theta}_t \right) \right\|^2 - \frac{\alpha}{2} \left\| \frac{1}{M} \sum_{m \in \mathcal{M}} \nabla \widetilde{F}^m \left( \boldsymbol{\theta}_t^m \right) \right\|^2 + \frac{\alpha}{2} \frac{1}{M} \sum_{m \in \mathcal{M}} \left\| \nabla F^m \left( \boldsymbol{\theta}_t^m \right) - \nabla \widetilde{F}^m \left( \boldsymbol{\theta}_t^m \right) \right\|^2$

$+ \frac{L_F}{2} \alpha^2 \left\| \frac{1}{M} \sum_{m \in \mathcal{M}} \nabla \widetilde{F}^m \left( \boldsymbol{\theta}_t^m \right) \right\|^2$

$\overset{(e)}{\le} -\frac{\alpha}{2} \left\| \nabla \mathcal{L} \left( \boldsymbol{\theta}_t \right) \right\|^2 + \frac{\alpha}{2} \frac{1}{M} \sum_{m \in \mathcal{M}} \left\| \nabla F^m \left( \boldsymbol{\theta}_t^m \right) - \nabla \widetilde{F}^m \left( \boldsymbol{\theta}_t^m \right) \right\|^2, \quad (15)$

where (a) follows from Lemma D.2, (b) is because of the update rule in Swift-FedGNN, (c) uses $\langle \boldsymbol{x}, \boldsymbol{y} \rangle = \frac{1}{2} \|\boldsymbol{x}\|^2 + \frac{1}{2} \|\boldsymbol{y}\|^2 - \frac{1}{2} \|\boldsymbol{x} - \boldsymbol{y}\|^2$, (d) utilizes $\left\| \sum_{i=1}^{n} \boldsymbol{x}_i \right\|^2 \le n \sum_{i=1}^{n} \|\boldsymbol{x}_i\|^2$, and (e) is due to the choice of $\alpha \le 1/L_F$.

When $t \in [(n_t - 1) I + 1, n_t I - 1] \cap \mathbb{Z}$, where $n_t = \{1, 2, \cdots\}$, Swift-FedGNN conducts local training for all clients $m \in \mathcal{M}$. Thus,

$$\left\| \nabla F^m \left( \boldsymbol{\theta}_t^m \right) - \nabla \widetilde{F}^m \left( \boldsymbol{\theta}_t^m \right) \right\| = \left\| \nabla F_{full}^m \left( \boldsymbol{\theta}_t^m \right) - \nabla \widetilde{F}_{local}^m \left( \boldsymbol{\theta}_t^m \right) \right\|$$

$$\le \left\| \nabla F_{full}^m \left( \boldsymbol{\theta}_t^m \right) - \nabla F_{local}^m \left( \boldsymbol{\theta}_t^m \right) \right\| + \left\| \nabla F_{local}^m \left( \boldsymbol{\theta}_t^m \right) - \nabla \widetilde{F}_{local}^m \left( \boldsymbol{\theta}_t^m \right) \right\|$$

$$\overset{(a)}{\le} LB^r_{\Delta G} + LB^l_{\Delta G}, \tag{16}$$

where (a) follows from Lemmas D.6 and D.7.

When $t = n_t I$, where $n_t = \{1, 2, \cdots\}$, Swift-FedGNN performs local training for clients $m \in \mathcal{M} \backslash \mathcal{K}$, and thus the inequality (16) holds for these clients. The randomly sampled clients $m \in \mathcal{K}$ conduct cross-client training, and thus

$$\left\| \nabla F^m \left( \boldsymbol{\theta}^m_t \right) - \nabla \widetilde{F}^m \left( \boldsymbol{\theta}^m_t \right) \right\| = \left\| \nabla F^m_{full} \left( \boldsymbol{\theta}^m_t \right) - \nabla \widetilde{F}^m_{full} \left( \boldsymbol{\theta}^m_t \right) \right\| \overset{(a)}{\le} LB^f_{\Delta G},$$

where (a) uses Lemma D.6.

Telescoping (15) from $i = (n_t - 1) I + 1$ to $n_t I$, we have

$$\sum_{i=(n_t-1)I+1}^{n_t I} \left( \mathcal{L} \left( \boldsymbol{\theta}_{i+1} \right) - \mathcal{L} \left( \boldsymbol{\theta}_i \right) \right)$$

$$\le -\frac{\alpha}{2} \sum_{i=(n_t-1)I+1}^{n_t I} \left\| \nabla \mathcal{L} \left( \boldsymbol{\theta}_i \right) \right\|^2 + \frac{\alpha}{2}(I-1)L^2 \left( B^l_{\Delta G} + B^r_{\Delta G} \right)^2 + \frac{\alpha}{2M} K L^2 \left( B^f_{\Delta G} \right)^2$$

$$+ \frac{\alpha}{2M}(M-K)L^2 \left( B^l_{\Delta G} + B^r_{\Delta G} \right)^2.$$

Choosing $T = n_t I$ yields

$$\sum_{t=0}^{T-1} \left( \mathcal{L} \left( \boldsymbol{\theta}_{t+1} \right) - \mathcal{L} \left( \boldsymbol{\theta}_t \right) \right)$$

$$\le -\frac{\alpha}{2} \sum_{t=0}^{T-1} \left\| \nabla \mathcal{L} \left( \boldsymbol{\theta}_t \right) \right\|^2 + \frac{\alpha}{2}(T - n_t)L^2 \left( B^l_{\Delta G} + B^r_{\Delta G} \right)^2 + n_t \frac{\alpha}{2M} K L^2 \left( B^f_{\Delta G} \right)^2$$

$$+ n_t \frac{\alpha}{2M}(M-K)L^2 \left( B^l_{\Delta G} + B^r_{\Delta G} \right)^2.$$

Rearranging the terms and multiplying both sides by $2/\alpha$, we get

$$\sum_{t=0}^{T-1} \left\| \nabla \mathcal{L} \left( \boldsymbol{\theta}_t \right) \right\|^2$$

$$\le \frac{2}{\alpha} \sum_{t=0}^{T-1} \left( \mathcal{L} \left( \boldsymbol{\theta}_t \right) - \mathcal{L} \left( \boldsymbol{\theta}_{t+1} \right) \right) + (T - n_t)L^2 \left( B^l_{\Delta G} + B^r_{\Delta G} \right)^2 + \frac{n_t}{M} K L^2 \left( B^f_{\Delta G} \right)^2$$

$$+ \frac{n_t}{M}(M-K)L^2 \left( B^l_{\Delta G} + B^r_{\Delta G} \right)^2.$$

Dividing both sides by $T$ and choosing $\alpha = \sqrt{M}/\sqrt{T}$ completes the proof of Theorem 5.6.

□

