# OpenReview forum: "Swift-FedGNN: Federated Graph Learning with Low Communication and Sample Complexities"
_ICLR.cc/2025/Conference — Submitted to ICLR 2025_

### Official Review · Reviewer_ezAA · 2024-11-01

**Soundness:** 2
**Presentation:** 3
**Contribution:** 2
**Rating:** 3
**Confidence:** 4

**Summary:**

The paper presents Swift-FedGNN, a federated learning framework for graph neural networks that reduces sampling and communication overhead. It achieves this by focusing on local training with periodic cross-client updates to handle geo-distributed data while preserving privacy. Key contributions include efficient federated GNN training with minimal cross-client communication and a convergence rate that matches state-of-the-art methods.

**Strengths:**

1. The paper clearly explains the methodologies.
2. Swift-FedGNN achieves strong convergence rates comparable to state-of-the-art methods.

**Weaknesses:**

1. In Section 4, the basis for neighbor sampling is unclear. How are nodes chosen, and how is it confirmed that they are true neighbors? Additionally, it’s uncertain how cross-client neighbors are verified between clients. Are there any specific assumptions regarding this?
2. The novelty primarily lies in local training with periodic cross-client updates, which may limit the contributions, as periodic updates are not a particularly novel technical innovation.
3. Comparing Swift-FedGNN with more recent baselines, as well as classic methods like FedAvg, FedSage, etc., would strengthen the study.
4. What is the primary factor contributing to Swift-FedGNN’s fast convergence rate? In Figure 4, the x-axis shows Wall-Clock Time. Would Swift-FedGNN still converge the fastest if the x-axis were set to iterations? I recommend including an additional plot of convergence vs. iterations, along with a more detailed analysis of the factors contributing to fast wall-clock time convergence.

**Questions:**

Please see and address my concerns on the weaknesses above.

---

> ### Author Response · Authors · 2024-11-24
>
> ### Weaknesses:
>
> > **Comment 1:** In Section 4, the basis for neighbor sampling is unclear. How are nodes chosen, and how is it confirmed that they are true neighbors? Additionally, it’s uncertain how cross-client neighbors are verified between clients. Are there any specific assumptions regarding this?
>
>
> **Our Response:** Thanks for your comment. Your "true neighbor" question can be resolved in the setting where i) the server has an accurate full graph topology information; and ii) all clients have accurate local subgraph information for the local graph sampling. These will ensure that the verification and management of cross-client and local neighbor relationships, guaranteeing that the sampled cross-client and local neighbors are indeed true neighbors.
>
>
> > **Comment 2:** The novelty primarily lies in local training with periodic cross-client updates, which may limit the contributions, as periodic updates are not a particularly novel technical innovation.
>
>
> **Our Response:** Thanks for your comment. While periodic cross-client updates are a component of our approach, we believe the novelty of our work extends far beyond this aspect. To the best of our knowledge, the only prior work leveraging periodic cross-client updates in federated GNNs is [Du & Wu, 2022]. However, our method significantly differs from theirs in the following three key aspects:
>
> 1. In [Du & Wu, 2022], sparse cross-client neighbor sampling is used to reduce communication overhead by periodically sampling cross-client neighbors and exchanging their information between clients. While this approach supplements lost cross-client neighbor information, it introduces additional memory overhead because clients must cache transferred graph data. Moreover, as training progresses, the frequency of information exchange increases, leading to higher communication costs.
> 2. [Du & Wu, 2022] relaxes privacy constraints by allowing direct transfer of graph data between clients, which raises privacy concerns. Additionally, reusing the same sampled data across multiple iterations can introduce bias, degrading the model’s performance.
> 3. In contrast, our proposed Swift-FedGNN method limits cross-client training to a subset of sampled clients and **avoids direct graph data exchange** by offloading certain operations to a central server. Cross-client neighbor information is aggregated in two stages: first at the remote clients and then at the server. This two-stage aggregation process preserves data privacy and significantly reduces communication costs. Furthermore, since local training occurs in the remaining iterations, clients do **not** need to store cross-client neighbor information, thereby reducing memory overhead.
>
> [Du & Wu, 2022] Bingqian Du and Chuan Wu. Federated graph learning with periodic neighbour sampling. In 2022 IEEE/ACM 30th International Symposium on Quality of Service (IWQoS), pp. 1–10. IEEE, 2022.
>
> > **Comment 3:** Comparing Swift-FedGNN with more recent baselines, as well as classic methods like FedAvg, FedSage, etc., would strengthen the study.
>
>
> **Our Response:** Thanks for your comment. We would like to clarify that traditional federated learning algorithms, such as FedAvg, are not directly applicable to federated GNN scenarios. This is primarily due to the inherent dependencies between nodes in a graph and the unique characteristics of real-world graph datasets, which are often extremely large and generated in a geo-distributed fashion. Unlike traditional datasets where data points are typically independent, nodes in graph data are highly interdependent. As a result, a node’s neighbors may be distributed across multiple clients, requiring additional mechanisms to handle cross-client neighbor information effectively.
>
> The baselines used in our experiments were chosen because they are the most closely related to our proposed algorithm. These baselines specifically address the challenges of reducing communication costs in federated GNNs and mitigating information loss from cross-client neighbors through periodic (sampling-based) full-neighbor training. By comparing against these targeted baselines, we ensure a more meaningful evaluation of our method's effectiveness.

---

> ### Author Response · Authors · 2024-11-24
>
> > **Comment 4:** What is the primary factor contributing to Swift-FedGNN’s fast convergence rate? In Figure 4, the x-axis shows Wall-Clock Time. Would Swift-FedGNN still converge the fastest if the x-axis were set to iterations? I recommend including an additional plot of convergence vs. iterations, along with a more detailed analysis of the factors contributing to fast wall-clock time convergence.
>
>
> **Our Response:** Thank you for your suggestion. The primary factor contributing to Swift-FedGNN’s fast convergence rate is its ability to **reduce the frequency of sampling and communication between clients**, which are *the most time-consuming aspects* of the process.
>
> To answer your question regarding "convergence vs. iterations," in this rebuttal period, we have generated additional plots to illustrate convergence vs iterations (see Figure 7 in Appendix C.1) in the revised version of our paper. The numbers of iterations required by all algorithms in comparison are similar, which is due to the fact that they share the same convergence rate result. However, since Swift-FedGNN minimizes the sampling and communication overhead, it achieves the lowest **wall-clock time** for convergence, making it the most efficient in terms of practical implementation.

---

> > ### Comment · Reviewer_ezAA · 2024-11-26
> > **Thanks for the rebuttal**
> >
> > Thank you for the rebuttal. However, the response does not fully address my concerns.
> >
> > 1. It is not realistic to assume that "i) the server has accurate full graph topology information; and ii) all clients have accurate local subgraph information for the local graph sampling" in real-world environments. While such assumptions can be made, they significantly limit the practical usefulness of the proposed framework.
> >
> > 2. I find the explanation for fast convergence unconvincing. The proposed method requires a similar number of communication rounds to converge compared to other methods, making it inappropriate to claim fast convergence.

---

> ### Author Response · Authors · 2024-11-27
>
> > **Comment 1:** It is not realistic to assume that "i) the server has accurate full graph topology information; and ii) all clients have accurate local subgraph information for the local graph sampling" in real-world environments. While such assumptions can be made, they significantly limit the practical usefulness of the proposed framework.
>
> **Our Response:** Thanks for your comments. While it may not always be realistic to assume that the server has accurate full graph topology information, this assumption is **far weaker** compared to those much stronger assumptions commonly used in federated GNN literature (e.g., clients have access to the features of cross-client neighbor nodes located on other clients [1]-[3]). By relaxing these strong assumptions of the existing approaches, our proposed algorithm has significantly broadened the scope of practical applications compared to existing federated GNN approaches.
>
> Regarding the assumption that clients have accurate local subgraph information, this is not unique to our framework but is a standard assumption in the federated GNN literature. This is because each client is responsible for generating and maintaining its own local subgraph, which inherently provides accurate information about its structure. We would appreciate it if the reviewer could provide some *specific* scenarios where clients do not have accurate local subgraph information, which will help us improve our work to address such situations.
>
> [1] Liu, Rui, Pengwei Xing, Zichao Deng, Anran Li, Cuntai Guan, and Han Yu. "Federated graph neural networks: Overview, techniques, and challenges." IEEE Transactions on Neural Networks and Learning Systems (2024).
>
> [2] Zhang, Ke, Carl Yang, Xiaoxiao Li, Lichao Sun, and Siu Ming Yiu. "Subgraph federated learning with missing neighbor generation." Advances in Neural Information Processing Systems 34 (2021): 6671-6682.
>
> [3] Du, Bingqian, and Chuan Wu. "Federated graph learning with periodic neighbour sampling." In 2022 IEEE/ACM 30th International Symposium on Quality of Service (IWQoS), pp. 1-10. IEEE, 2022.
>
>
>
>
> > **Comment 2:** I find the explanation for fast convergence unconvincing. The proposed method requires a similar number of communication rounds to converge compared to other methods, making it inappropriate to claim fast convergence.
>
> **Our Response:** Thank you for your comment. We would like to clarify that the claim of "fast convergence speed" in our Numerical Results section specifically refers to **reduced wall-clock time**, *not* the number of communication rounds. While our proposed algorithm requires a similar number of training rounds compared to the baselines, Swift-FedGNN achieves **much shorter wall-clock time** due to its lower cross-client sampling and communication requirements, which are the *most time-consuming* operations in federated GNNs.
>
> Additionally, we would like to point out that our proposed algorithm **does not require a similar number of communication rounds** to converge compared to other methods. Instead, it requires a similar number of **training rounds** but **much fewer communication rounds** (**Note:** training round and communications round are two completely different notions. In our method, a communication round contains multiple training rounds). The similarity in the numbmer of training rounds is expected, as the theoretical convergence rate achieved by our algorithm matches with that of state-of-the-art baseline algorithms. Furthermore, the number of communication rounds required by our algorithm is given by $T/I$, where $T$ is the total number of training rounds and $I$ is the communication frequency.
>
> It is important to note that comparing the number of communication rounds directly across different algorithms can be misleading, as methods with fewer communication rounds may require a larger amount of data to be exchanged per round. Therefore, it is more appropriate to evaluate the overall communication efficiency and wall-clock time rather than focusing solely on the number of communication rounds.

---

### Official Review · Reviewer_xG4w · 2024-11-01

**Soundness:** 3
**Presentation:** 3
**Contribution:** 3
**Rating:** 6
**Confidence:** 3

**Summary:**

1. The paper proposes a technique for federated learning in graph neural networks.
2. Thorough mathematical analysis shows that using this method reduces memory overhead.
3. Starting from mild assumptions the authors provide competitive convergence rate guarantees.
4. Some empirical evidence is presented to support theoretical results.

**Strengths:**

1. The theoretical derivations have clear assumptions that do not seem excessive. Bounded gradients (Lipschitz) is a mild assumption in general. Though the paper plus appendix are long and quote-dense (30 pages total) I believe that the theoretical derivations are correct and the claims are validated. Disclaimer: I read the proofs until page 23, Lemma D.5. All seemed correct. Glancing at the rest due to limited time for the review I believe that they are correct though I have not checked the details for the last 8 pages of the appendix.

2. The method does what it claims: it provides a new communication and sample-efficient mini-batch algorithm for Federated learning. It appears to outperform comparable SOTA approaches in terms of convergence.

3. Convincing experimental results. Though it is only tested on two datasets the algorithm is appears to have a clear edge, e.g., in Figure 4.

**Weaknesses:**

1. While the theoretical grounding is excellent, the experiments seem a little too narrow. The use of only 2 benchmarks raises doubts as to if the relative improvement to baselines is generalizable beyond the selected datasets. Can the authors explain the choice of datasets? Why not use other datasets that are in baselines such as Flickr, OGB-Products, OGB-Arxiv?.I suggest including these datasets for experiments.

2. We do not have hard metrics. How do the different methods perform at the end of training in the test set? Just because the validation loss converges faster in Fig. 4 does not mean that the test metric at the end of training will be better. Please provide a table comparing final test set performance across methods using standard metrics like accuracy, ROC-AUC, and F1 score

3. The paper is quite dense, I recommend rewriting parts of the introduction, contributions, and review to shorten them or to move part of that text to the appendix to make the text less dense. Some figures are hard to read, Figures 4,5,6 require zooming to 200% or more to be able to read clearly and this disrupts the flow of the paper.

**Questions:**

See questions in weaknesses. Additional questions:

1. Reported memory usage in Table 1 of LLCG (https://openreview.net/pdf?id=FndDxSz3LxQ) reports different Megabytes of node representation/feature communicated per round in Reddit than Table 2 of your paper. Why is that? Is there a way to make sure there is an apples to apples comparison here?.

2. It seems a little troubling that the baselines are 2-3 years old. Are there really no newer methods to compare to? It seems a little concerning if research interest in this setup is not very high from an impact perspective. Can the authors alleviate my concerns here? If not, please add and compare with recent state-of-the-art to strengthen the paper.

I will raise it to 6 if they can answer my questions about datasets and why so few baselines as to how it justifies motivation.

---

> ### Author Response · Authors · 2024-11-24
>
> ### Weaknesses:
>
> > **Comment 1:** While the theoretical grounding is excellent, the experiments seem a little too narrow. The use of only 2 benchmarks raises doubts as to if the relative improvement to baselines is generalizable beyond the selected datasets. Can the authors explain the choice of datasets? Why not use other datasets that are in baselines such as Flickr, OGB-Products, OGB-Arxiv?.I suggest including these datasets for experiments.
>
>
> **Our Response:** Thank you for your comment. We selected the ogbn-products dataset because it is already the **largest dataset** commonly used in the federated GNN literature, providing a comprehensive benchmark for scalability. Similarly, the Reddit dataset was chosen due to its **high density**, with an average of approximately 600 neighbors per node, which makes it well-suited for evaluating the performance of GNNs in dense graph settings. These datasets were specifically chosen for their distinct and representative characteristics to test the versatility of our proposed framework.
>
> We would like to clarify that our paper already includes experiments on the OGB-Products dataset, which we refer to as Ogbn-products in our paper. We assume your suggestion may have been intended to refer to OGB-Proteins instead. Regarding datasets such as Flickr, OGB-Proteins, and OGB-Arxiv, we chose not to include them *because they are relatively small graphs, with fewer than 0.17 million nodes*. These smaller datasets are less suitable for evaluating the scalability and effectiveness of our proposed framework, which is designed to handle larger and more complex graphs.
>
> We appreciate your suggestion and understand the importance of evaluating additional datasets. We are currently working on these experiments and will include them in the revised version of our paper once they are completed.
>
>
> > **Comment 2:** We do not have hard metrics. How do the different methods perform at the end of training in the test set? Just because the validation loss converges faster in Fig. 4 does not mean that the test metric at the end of training will be better. Please provide a table comparing final test set performance across methods using standard metrics like accuracy, ROC-AUC, and F1 score
>
>
> **Our Response:** Thank you for your comment. In this rebuttal period, we have been able to conduct additional experiments to provide the final validation accuracy, which are summarized in Tables 5 and 6 and demonstrate that our proposed algorithm achieves accuracy comparable to state-of-the-art algorithms while significantly reducing communication and sampling costs.
>
> It is worth noting that FedGNN-G achieves slightly better accuracy, as it is trained on the *full graph*  without the loss of cross-client neighbor information (hence a somewhat unfair comparison). However, our approach **balances accuracy with efficiency**, making it more practical for real-world federated GNN scenarios.
>
>
> Table 5: Validation Accuracy with Baselines on ogbn-products
> | Method | Swift-FedGNN |  FedGNN-PNS | LLCG | FedGNN-G |
> | -------- | -------- | -------- | -------- |  -------- |
> | Validation Accuracy (%)     |  90.00    |   90.20   | 89.50 | 91.55   |
>
> Table 6: Validation Accuracy with Baselines on reddit
> | Method | Swift-FedGNN | FedGNN-PNS | LLCG | FedGNN-G |
> | -------- | -------- | -------- | -------- |  -------- |
> | Validation Accuracy (%)     |  95.90    |   96.25   | 95.65 | 97.10    |
>
> > **Comment 3:** The paper is quite dense, I recommend rewriting parts of the introduction, contributions, and review to shorten them or to move part of that text to the appendix to make the text less dense. Some figures are hard to read, Figures 4,5,6 require zooming to 200% or more to be able to read clearly and this disrupts the flow of the paper.
>
>
> **Our Response:** Thanks for your suggestions. We will revise the introduction, contributions, and related work sections according to your suggestions to make the text more concise, and we will move some detailed contents to the appendix to improve readability. Additionally, we will update Figures 4, 5, and 6 to enhance their clarity and ensure they are easily readable without requiring significant zooming. These adjustments will be included in the revised version of our paper.

---

> ### Author Response · Authors · 2024-11-24
>
> ### Questions:
>
> > **Comment 1:** Reported memory usage in Table 1 of LLCG (https://openreview.net/pdf?id=FndDxSz3LxQ) reports different Megabytes of node representation/feature communicated per round in Reddit than Table 2 of your paper. Why is that? Is there a way to make sure there is an apples to apples comparison here?.
>
>
> **Our Response:** Thanks for your questions. The differences in reported memory usage between Table 1 of LLCG and Table 2 of our paper stem from variations in experimental configurations, such as the number of clients, batch size, and the number of neighbors sampled per round. These factors can significantly impact the average Megabytes of communication per round. In addition, random sampling can also affect the communication cost.
>
> To ensure a fair comparison in our study, we apply consistent experimental configurations across all baselines evaluated in our paper. This approach allows us to compare the methods under identical conditions, ensuring an apples-to-apples comparison within our experiments.
>
> *We stand by our results and will open-source all our code, datasets, and experiment settings for the research community to reproduce our results and verify our findings.*
>
>
> > **Comment 2:** It seems a little troubling that the baselines are 2-3 years old. Are there really no newer methods to compare to? It seems a little concerning if research interest in this setup is not very high from an impact perspective. Can the authors alleviate my concerns here? If not, please add and compare with recent state-of-the-art to strengthen the paper.
>
>
> **Our Response:** Thanks for your comment. The baselines used in our experiments were chosen because they are the most closely related to our proposed algorithm. These baselines specifically address the challenges of reducing communication costs in federated GNNs and mitigating information loss from cross-client neighbors through periodic (sampling-based) full-neighbor training.
>
> While it is true that the baselines are 2–3 years old, they remain representative of state-of-the-art approaches in the federated GNN domain and are widely adopted in current research. By focusing on these targeted baselines, we ensure a meaningful and relevant evaluation of our method's effectiveness.
>
> The ages of these baseline methods do **not necessarily mean** a lack of research interest in this area. Instead, it reflects the **fundamental hardness** of the problem of communication-efficient federated GNN training compared to problems in the ML literature. As a result, making significant breakthroughs to advance the state of the art is challenging. Furthermore, the **pressing need** to run large GNNs in a distributed manner underscores the importance of addressing the communication challenges in federated GNNs. In fact, *the longer these challenges persist, the more urgent it becomes for the research community to tackle them.*

---

> > ### Comment · Reviewer_xG4w · 2024-12-01
> > **Response to Reviewers**
> >
> > Thanks for the detailed clarification. Although I am not convinced by a few responses (Weaknesses, Comment 2 and Questions, Comment 2), I value the method and I believe the proposed idea is beneficial. Also, authors need to make more changes to make the paper flow look easy. The unconventional approach of making the paper appear overly dense may deter readers from engaging with it. I hope the authors will make these changes. I will raise my score to 6.

---

### Official Review · Reviewer_Byak · 2024-11-03

**Soundness:** 3
**Presentation:** 4
**Contribution:** 3
**Rating:** 5
**Confidence:** 4

**Summary:**

This paper focuses on federated graph neural network training. Specifically, the clients share the same set of GNN model parameters, and each client holds a subgraph of the graph dataset (with partial non-overlapping nodes). The proposed algorithm aims to overcome the heavy communication burden in subgraph sampling for GNN training through 1) periodical inter-client training and multiple steps of local training and 2) client sampling for inter-client training. The authors provided a theoretical convergence analysis of the proposed algorithm with $O(1/\sqrt{T})$ rate. Experiments on the OGBN and Reddit datasets show that the proposed algorithm outperforms existing FL GNN algorithms under the same settings.

**Strengths:**

1. Originality: This paper provides a novel algorithm for (vertical) Federated Learning on Graph data. The algorithm analysis is also novel compared with existing FL and GNN algorithms.
2. Clarity: The paper is clearly written with rigorous definitions of the notations and assumptions, problem statements, algorithm descriptions, and experiment settings.

**Weaknesses:**

1. Missing ablation study: The numerical results cannot fully support its theoretical outcomes. The authors failed to conduct sufficient ablation study, including:
    1. The effect of $K, I$ on the final accuracy since they affect the residual error.
    2. Estimating the values of $B^f, B^l, B^r$ on how they changes with $L,B$'s in the assumptions.
    3. The experiments on models other than GraphSAGE, e.g., on GCN and GAT.
    4. Experiments on more datasets, e.g., other OGBN datasets, are also desirable.
2. Inaccurate theoretical statement:
    1. The claim of $O(1/\sqrt{T})$ convergence is inaccurate since Thm. 5.6 indicates that the algorithm only converges to a neighborhood of the exact solution.
    2. Further discussion on the convergence result compared with existing algorithms is missing. Specifically, how the residual error compares with SOTA algorithms, e.g.,  Chen et al., 2018; Cong et al., 2021; Ramezani et al., 2022; and Du & Wu, 2022.
    3. The remark (3) in lines 448-449 is not proved. Please refer to the place where this remark is proven.
3. Missing privacy analysis & statement
    The author claimed that aggregating the node embedding of layer $L$ is more private than communicating the node features. The authors should provide a more rigorous discussion on this claim, e.g., what privacy is protected (node feature? edge?) and how they link to real applications such as hospitals; how much "more" privacy is protected with the proposed algorithm?

**Questions:**

See the weaknesses above.

Specifically, please address weaknesses 1.1, 1.3, 2.2, 2.3

---

> ### Author Response · Authors · 2024-11-24
>
> ### Weaknesses:
>
> > **Comment 1:** Missing ablation study: The numerical results cannot fully support its theoretical outcomes. The authors failed to conduct sufficient ablation study, including:
> > 1. The effect of $K,I$ on the final accuracy since they affect the residual error.
> > 2. Estimating the values of $B^f, B^l, B^r$ on how they changes with $L,B$'s in the assumptions.
> > 3. The experiments on models other than GraphSAGE, e.g., on GCN and GAT.
> > 4. Experiments on more datasets, e.g., other OGBN datasets, are also desirable.
>
>
> **Our Response:** Thanks for your comments. In this rebuttal period, we have managed to conduct new ablation studies as you suggested. Please see Tables 1 and 2 as well as our point-to-point responses below:
>
> 1. We report the validation accuracy for different correction frequencies ($I$) and numbers of sampled clients ($K$) in Tables 1 and 2, respectively. The results demonstrate that as $I$ increases, validation accuracy decreases, while increasing $K$ leads to improved validation accuracy. These findings are consistent with our theoretical conclusion in Remark (3) in Line 448 on Page 9.
> 2. We would like to clarify that $L$ and $B$s are constants determined by the dataset and problem instance. These values are inherent properties of the data and are not influenced by the algorithm’s design or components. As such, they do not behave like algorithm hyperparameters that can be adjusted or studied through ablation analysis. Consequently, for most papers on optmization algorithm designs in the areas of machine learning, it is often difficulty, if not entirely not possible, to conduct an ablation study on parameters $L$ and $B$s in the same way as for other tunable components of the algorithm.
> 3. GraphSAGE is one of the most commonly used sampling-based models in the federated GNN literature, which is why we selected it for our experiments. We would like to clarify that our Swift-FedGNN framework does not currently support GAT, as GAT involves non-element-wise operations, whereas our framework offloads certain operations to the server. Supporting GAT would require additional modifications to our method, which is highly non-trivial and warrants a dedicated paper to this design. Furthermore, GCN is a *transductive* algorithm that relies on the full graph structure during training. This makes it **unsuitable** for our sampling-based Swift-FedGNN framework, which is designed to work in scenarios where the entire graph structure is not available.
> 4. We would like to clarify that the datasets used in our experiments were carefully selected for their distinct and representative characteristics. Specifically, the ogbn-products dataset is the largest dataset commonly used in the federated GNN literature, making it a comprehensive benchmark for evaluating scalability. Similarly, the Reddit dataset is well-known for its high density, with an average of approximately 600 neighbors per node, making it ideal for assessing the performance of GNNs in dense graph settings. We appreciate your suggestion and agree that additional experiments on other datasets, such as other OGBN datasets, could further strengthen our study. We are currently working on these experiments and will include them in the revised version of our paper once they are completed.
>
>
> Table 1: Validation Accuracy with Different Correction Frequency ($I$)
> | Correction Frequencies ($I$) | 5 | 20 | 40 |
> | -------- | -------- | -------- |-------- |
> | Validation Accuracy (%)     | 89.95     | 89.85     | 89.80 |
>
>
>
> Table 2: Validation Accuracy with Different Number of Sampled Clients ($K$)
> | # of Sampled Clients ($K$) | 1 | 10 | 15|
> | -------- | -------- | -------- |-------- |
> | Validation Accuracy (%)     | 89.75     | 89.80     | 89.90 |

---

> ### Author Response · Authors · 2024-11-24
>
> > **Comment 2:** Inaccurate theoretical statement:
> > 1. The claim of $O(1/\sqrt{T})$ convergence is inaccurate since Thm. 5.6 indicates that the algorithm only converges to a neighborhood of the exact solution.
> > 2. Further discussion on the convergence result compared with existing algorithms is missing. Specifically, how the residual error compares with SOTA algorithms, e.g., Chen et al., 2018; Cong et al., 2021; Ramezani et al., 2022; and Du & Wu, 2022.
> > 3. The remark (3) in lines 448-449 is not proved. Please refer to the place where this remark is proven.
>
>
> **Our Response:** Thank you for your comments. Our point-to-point responses to your comments are as follows:
>
> 1. We agree that the convergence rate of Swift-FedGNN is $O\left( 1/\sqrt{T} \right)$ to a neighborhood of the exact solution, as indicated in Theorem 5.6. We appreciate your observation and have clarified this distinction in the revised version of our paper.
> 2. Comparing residual errors across different SOTA algorithms is challenging due to varying assumptions. However, our experimental results (see Tables 3 and 4 below) demonstrate that our proposed framework achieves comparable accuracy to SOTA algorithms while significantly reducing sampling and communication costs. We did not compare our algorithm with those of Chen et al., 2018 and Cong et al., 2021, since both of them rely on variance reduction techniques to enhance performance. Meanwhile, these algorithms will also inherent all the computation difficulties of VR-based methods (e.g., double-loop structure, periodic full gradient computation), which are absent in our proposed method. Hence, these algorithms are not a fair comparisons to ours. On the other hand, our proposed algorithm can also be equipped with variance reduction methods to further improve its performance, making it adaptable to such enhancements if needed.
> 3. To support Remark (3) in lines 448–449, we define $B_{\Delta P}= \max \left\\{ B_{\Delta P}^l, B_{\Delta P}^f \right\\}$ and $B_{\Delta X}= \max \left\\{ B_{\Delta X}^l, B_{\Delta X}^f \right\\}$. Based on Lemmas D.3, D.4 and D6, we prove that $B_{\Delta G}^l=B_{\Delta G}^f$. Thus, we can establish that $B_{\Delta G}^f \lt B_{\Delta G}^l+B_{\Delta G}^r$. Consequently, the last term of the convergence bound in Theorem 5.6 is negative. We will add the proof of this remark in the revised version of our paper.
>
> Table 3: Validation Accuracy with Baselines on ogbn-products dataset
> | Method | Swift-FedGNN | FedGNN-PNS (Du & Wu, 2022) | LLCG (Ramezani et al., 2022) |
> | -------- | -------- | -------- | -------- |
> | Validation Accuracy (%)     |  90.00      |  90.20   | 89.50 |
>
> Table 4: Validation Accuracy with Baselines on reddit dataset
> | Method | Swift-FedGNN | FedGNN-PNS (Du & Wu, 2022) | LLCG (Ramezani et al., 2022) |
> | -------- | -------- | -------- | -------- |
> | Validation Accuracy (%)     |  95.90     |  96.25   | 95.65 |

---

> ### Author Response · Authors · 2024-11-24
>
> > **Comment 3:** Missing privacy analysis & statement The author claimed that aggregating the node embedding of layer $L$ is more private than communicating the node features. The authors should provide a more rigorous discussion on this claim, e.g., what privacy is protected (node feature? edge?) and how they link to real applications such as hospitals; how much "more" privacy is protected with the proposed algorithm?
>
>
> **Our Response:** Thank you for your comment. Our proposed aggregation mechanism provides privacy protection for **both node features and edges**. Specifically, it ensures that clients cannot access the individual features of nodes, the number of nodes held by other clients, or the explicit connections between nodes. Additionally, by performing aggregation on the server, clients are unable to determine the locations of their cross-client neighbors. To see this, consider the following concrete example.
>
> *Example:* consider a scenario involving hospitals during the COVID-19 pandemic. Suppose Patient 1 is located at Hospital A and has connections (e.g., interactions or shared characteristics) with individuals at Hospitals B and C. Under our algorithm, Hospital A does not gain access to the individual patient information or connection details from Hospitals B and C. Instead, Hospitals B and C first locally aggregate the features of Patient 1's neighbors located on their respective premises and send the aggregated results to the server. The server then performs a second aggregation step before sending the final aggregated neighbor information back to Hospital A. This process safeguards privacy by ensuring that only aggregated neighbor information is shared, protecting sensitive data at both the node feature and edge levels.
>
> We note that privacy protection is an additional benefit of our algorithm. However, since privacy is not the primary focus of our algorithm, the design goal of our algorithm is not to optimize some quantifiable privacy metric and provide theoretical privacy guarantee (e.g., different privacy). We agree that federated GNN algorithms with theoretical and quantifiable privacy guarantee is an important direction and deserves an independent paper dedicated to this area. We will pursue this direction in our future studies.

---

### Official Review · Reviewer_uWRg · 2024-11-06

**Soundness:** 2
**Presentation:** 2
**Contribution:** 2
**Rating:** 5
**Confidence:** 4

**Summary:**

This paper proposes Swift-FedGNN. It aims to reduce the communication and sampling cost in federated graph learning. The framework is based on mini-batch training of GNNs. Clients in Swift-FedGNN do local training in most of time and periodically conduct cross-client training. The authors provide convergence theory of Swift-FedGNN.

**Strengths:**

S1: Writing is good.

S2: Related works are comprehensive.

S3: The results about the relationship between approximation errors and structure of GNNs are somewhat insightful.

**Weaknesses:**

W1: This paper claims to reduce the sampling and communication overhead, but comparison between Swift-FedGNN and most recent framework, e.g., FedGCN and FedSage+, is missing. It looks like FedGCN should has lower communication cost?

W2: The contribution seems a bit limited. Swift-FedGNN looks like a combination of FedGCN and [1]. I am not very sure about the novelty, I will check other reviewers’ comments.

W3: The authors remove Homomorphic Encryption in FedGCN, and claim “send only the aggregated results…help
preserve data privacy”. But according to FedGCN, the Homomorphic Encryption seems necessary for security.

Minor issues: In line 104 to 109, it seems like a statement is repeated twice?

[1] Federated graph learning with periodic neighbor sampling.

**Questions:**

Q1: In abstract and introduction, the authors motivate the federated graph learning with geo-distributed input graphs. But in experiments, graph partition is used for create multiple subgraphs. Is that a common practice? How do we know the subgraphs are distributed according to geo-information?

Q2: In line 182, the authors state “Note that this one-time communication only works for full graph training and is not suitable for the situations in  which clients sample different mini-batches of training nodes in each iteration since the cross-client neighbors of these training batches are different. ” I am not fully understand this, why the cross-client neighbors of these training batches are different? Can’t we just sample a training batch and acquire the aggregated representation of all their cross-neighbors?

---

> ### Author Response · Authors · 2024-11-24
>
> ### Weaknesses:
>
> > **Comment W1:** This paper claims to reduce the sampling and communication overhead, but comparison between Swift-FedGNN and most recent framework, e.g., FedGCN and FedSage+, is missing. It looks like FedGCN should has lower communication cost?
>
>
> **Our Response:** Thank you for your comment. We have discussed these comparisons in the "Related Work" section of our paper. Specifically, FedSage+ requires cross-client neighbor information exchange during each training round, resulting in substantial communication overhead.
>
> Regarding FedGCN, while it appears to have lower communication costs due to only requiring communication at the beginning of training, it necessitates the exchange of full cross-client neighbor information upfront. This approach is restricted to full-graph training, which is often impractical due to the immense size of real-world graphs. Additionally, FedGCN employs homomorphic encryption for privacy preservation, significantly increasing the communication overhead—by at least several times higher than communication without homomorphic encryption.
>
> It is worth noting that in practice, stochastic node and neighbor sampling techniques are commonly employed to manage large graphs. However, these techniques cannot be directly applied to FedGCN because the sampled training nodes differ in each training round, resulting in varying corresponding neighbors. Consequently, this necessitates repeated neighbor information exchanges in each training round, leading to substantial communication costs. These costs are further exacerbated by high computational costs due to the use of homomorphic encryption.
>
>
> > **Comment W2:** The contribution seems a bit limited. Swift-FedGNN looks like a combination of FedGCN and [1]. I am not very sure about the novelty, I will check other reviewers’ comments.
>
>
> **Our Response:** Thanks for your comment. We would like to clarify that Swift-FedGNN is **far from** a mere combination of FedGCN and [1], but rather a novel approach designed to address the limitations of both methods.
>
> Specifically, FedGCN cannot be directly applied in scenarios where the full graph too large to fit into each client device. To adapt FedGCN to such settings, cross-client neighbor information exchange and homomorphic encryption are required in every training round, resulting in both significant computational and communication costs.
>
> The method proposed in [1] reuses the sampled training nodes and their sampled cross-client neighbors across multiple training rounds. While this reduces the frequency of communication, it introduces additional memory overhead for caching the transferred graph data and causes bias in the training process, ultimately degrading model performance. Furthermore, as training progresses, the frequency of communication increases, leading to escalating communication costs. Additionally, the method in [1] relaxes privacy constraints by allowing direct transfer of graph data between clients, which raises significant privacy concerns.
>
> In contrast, our Swift-FedGNN method successfully addresses these challenges. Our method limits cross-client training to a subset of sampled clients and avoids direct graph data exchange between clients by offloading certain operations to the central server. Before communicating with the training clients, cross-client neighbor information is aggregated in two stages: first at the remote clients and then at the server. This approach helps preserve data privacy and significantly reduces communication costs. Moreover, since local training is performed by clients in the remaining iterations, cross-client neighbor information does **not** need to be stored, effectively reducing memory overhead.

---

> ### Author Response · Authors · 2024-11-24
>
> > **Comment W3:** The authors remove Homomorphic Encryption in FedGCN, and claim “send only the aggregated results…help preserve data privacy”. But according to FedGCN, the Homomorphic Encryption seems necessary for security.
>
>
> **Our Response:** Thanks for your comments. However, it is somewhat unclear to us what the word "security" the reviwer is specifically referring to in here. Note that the word "security" was never mentioned in the FedGCN paper in [Ref]. If the reviewer meant the "privacy" aspect of Homomorphic Encryption (HE) in FedGCN (note that the FedGCN paper only mentioned that *"we leverage homomorphic encryption (HE), which can preserve client privacy in federated learning but introduces significant overhead for each communication round"*), then we agree that HE can be used to preserve data privacy, as it ensures that clients cannot access the features of cross-client neighbors. However, we note that HE is **not** strictly necessary for federated GNNs, as privacy can **also** be preserved using alternative methods. Moreover, HE comes with significant computational and communication costs.
>
> In contrast, our proposed Swift-FedGNN adopts a two-stage aggregation process to help preserve data privacy without the high overhead associated with HE. The first aggregation occurs at the remote clients, and the second takes place on the central server. This approach ensures that clients still cannot access the features of cross-client neighbors, while also significantly reducing communication costs.
>
> It is important to note that if computational and communication costs are not a primary concern, HE could also be incorporated into Swift-FedGNN to further enhance data privacy. This flexibility allows our method to adapt to a variety of privacy and performance requirements.
>
> [Ref] Yuhang Yao, Weizhao Jin, Srivatsan Ravi, and Carlee Joe-Wong. FedGCN: Convergence-communication tradeoffs in federated training of graph convolutional networks. Advances in Neural Information Processing Systems, 36, 2023a.
>
> > **Comment:** Minor issues: In line 104 to 109, it seems like a statement is repeated twice?
>
>
> **Our Response:** Thank you for catching this typo. We have corrected the repetition in the revised version of our paper.
>
> ### Questions:
>
> > **Comment Q1:** In abstract and introduction, the authors motivate the federated graph learning with geo-distributed input graphs. But in experiments, graph partition is used for create multiple subgraphs. Is that a common practice? How do we know the subgraphs are distributed according to geo-information?
>
>
> **Our Response:** Thanks for your questions. It appears that there may be a misunderstanding of the word "geo-distributed." What we actually mean is the following: 1) the clients in a distributed federated learning system are located at different geo-locations; 2) each client at each different geo-location collects its own graph data; 3) collectively, the graph data at all clients form a global graph data (i.e., the graph data at each client is a subgraph of the glboal graph data).
>
> We note that the graph partition in our experiments is only an approach to simulate the aforementioned "geo-distributed" scenario. Using graph partitioning to create multiple subgraphs is a common practice for simulations in the Federated GNN literature, e.g., [1-5]. This approach is often employed to simulate the distributed nature of graph data across multiple clients.
>
> [1] Chaoyang He, Keshav Balasubramanian, Emir Ceyani, Carl Yang, Han Xie, Lichao Sun, Lifang He, Liangwei Yang, Philip S Yu, Yu Rong, et al. Fedgraphnn: A federated learning system and benchmark for graph neural networks. arXiv preprint arXiv:2104.07145, 2021a.
>
> [2] Chuhan Wu, Fangzhao Wu, Yang Cao, Yongfeng Huang, and Xing Xie. Fedgnn: Federated graph neural network for privacy-preserving recommendation. arXiv preprint arXiv:2102.04925, 2021.
>
> [3] Ke Zhang, Carl Yang, Xiaoxiao Li, Lichao Sun, and Siu Ming Yiu. Subgraph federated learning with missing neighbor generation. Advances in Neural Information Processing Systems, 34:6671–6682, 2021.
>
> [4] Yuhang Yao, Weizhao Jin, Srivatsan Ravi, and Carlee Joe-Wong. FedGCN: Convergence-communication tradeoffs in federated training of graph convolutional networks. Advances in Neural Information Processing Systems, 36, 2023a.
>
> [5] Bingqian Du and Chuan Wu. Federated graph learning with periodic neighbour sampling. In 2022 IEEE/ACM 30th International Symposium on Quality of Service (IWQoS), pp. 1–10. IEEE, 2022.

---

> ### Author Response · Authors · 2024-11-24
>
> > **Comment Q2:** In line 182, the authors state “Note that this one-time communication only works for full graph training and is not suitable for the situations in which clients sample different mini-batches of training nodes in each iteration since the cross-client neighbors of these training batches are different. ” I am not fully understand this, why the cross-client neighbors of these training batches are different? Can’t we just sample a training batch and acquire the aggregated representation of all their cross-neighbors?
>
>
> **Our Response:** Thank you for your question. In sampling-based Federated GNN algorithms, clients sample a new training batch in each training round. Each batch has its own corresponding neighbors, which can be located either locally or on other clients. To train effectively, the information for these cross-client neighbors must be acquired from the respective clients.
>
> Since the training batches change across rounds, the cross-client neighbors associated with these batches also differ. Consequently, communication of cross-client neighbor information is necessary for every training round. To better clarify this, consider the following toy example (see the table below):
> * Training Round 1: Client 1 samples nodes 1 and 2 as the training batch. The cross-client neighbors are nodes G and H (for node 2).
> * Training Round 2: Client 1 samples nodes 3 and 4 as the training batch. The cross-client neighbors are nodes I, J, and K (for node 4).
>
> This demonstrates that the cross-client neighbors of these training batches are different.
>
> Using the same training batch repeatedly throughout training could indeed eliminate the need for additional communication, but this approach would introduce bias, negatively impacting the model's performance and generalization ability. Therefore, dynamic sampling and corresponding cross-client communication are essential for ensuring both efficiency and performance in Federated GNNs.
>
>
> Table: Toy example of sampling in each training round
> | Training Round | 1 |  | 2 |  |
> | -------- | -------- | -------- |-------- | -------- |
> | Training Batch (Nodes)    | 1     | 2     | 3     | 4     |
> |1-Hop Local Neighbors |A,B |C|D|E,F|
> |1-Hop Cross-client Neighbors | |**G,H**||**I,J,K**|

---

### Meta-Review · Area_Chair_z6UK · 2024-12-25

**Metareview:**

The paper introduces a new mini-batch algorithm for federated learning over graph data that improves communication and sample efficiency -- local training with periodic cross-client updates. Several limitations were identified by the reviewers. The experimental evaluation was limited to only two datasets, and there were insufficient comparison metrics, notably missing accuracy, ROC-AUC, and F1 scores. The reviewers also questioned the novelty of the work from an algorithmic standpoint. Additionally, the paper relied on outdated baselines rather than recent methods, and there were some presentation issues including hard-to-read figures.

The reviewers provided several suggestions for improvement. They recommended testing on additional datasets such as Flickr and OGB, adding more comprehensive performance metrics, comparing against newer baselines, and clarifying memory usage discrepancies with prior work.

Overall, while the paper shows promise theoretically, it needs broader experimental validation and clearer demonstration of its novel contributions to strengthen its impact.

**Additional Comments On Reviewer Discussion:**

One of the reviewers suggested rasing the core but not convinced by responses.

---

### Decision · Program_Chairs · 2025-01-22

Reject